# Flexible auditory training, psychophysics, and enrichment of common marmosets with an automated, touchscreen-based system

A. Calapai [1,2,3,4,9], J. Cabrera-Moreno [2,3,5,6,9], T. Moser [3,5,6,7,8] & M. Jeschke [2,3,4,5 ✉]

Devising new and more efficient protocols to analyze the phenotypes of non-human primates, as well as their complex nervous systems, is rapidly becoming of paramount importance. This is because with genome-editing techniques, recently adopted to non-human primates, new animal models for fundamental and translational research have been established. One aspect in particular, namely cognitive hearing, has been difficult to assess compared to visual cognition. To address this, we devised autonomous, standardized, and unsupervised training and testing of auditory capabilities of common marmosets with a cage-based standalone, wireless system. All marmosets tested voluntarily operated the device on a daily basis and went from naïve to experienced at their own pace and with ease. Through a series of experiments, here we show, that animals autonomously learn to associate sounds with images; to flexibly discriminate sounds, and to detect sounds of varying loudness. The developed platform and training principles combine in-cage training of common marmosets for cognitive and psychoacoustic assessment with an enriched environment that does not rely on dietary restriction or social separation, in compliance with the 3Rs principle.

[1] Cognitive Neuroscience Laboratory, German Primate Center - Leibniz-Institute for Primate Research, Göttingen, Germany. [2] Cognitive Hearing in Primates (CHiP) Group, Auditory Neuroscience and Optogenetics Laboratory, German Primate Center - Leibniz-Institute for Primate Research, Göttingen, Germany. [3] Auditory Neuroscience and Optogenetics Laboratory, German Primate Center - Leibniz-Institute for Primate Research, Göttingen, Germany. [4] Leibniz ScienceCampus "Primate Cognition", Göttingen, Germany. [5] Institute for Auditory Neuroscience and InnerEarLab, University Medical Center Göttingen, 37075 Göttingen, Germany. [6] Göttingen Graduate School for Neurosciences, Biophysics and Molecular Biosciences, University of Göttingen, 37075 Göttingen, Germany. [7] Auditory Neuroscience Group and Synaptic Nanophysiology Group, Max Planck Institute for Multidisciplinary Sciences, 37077 Göttingen, Germany. [8] Cluster of Excellence "Multiscale Bioimaging: from Molecular Machines to Networks of Excitable Cells" (MBExC), University of Göttingen, 37075 Göttingen, Germany. [9] These authors contributed equally: A. Calapai, J. Cabrera-Moreno. ✉email: mjeschke@dpz.eu

In recent years non-human primates (NHP)s have seen increased interest as animal models for human diseases due to the advent of transgenic primates and genome-editing technologies[1,2]. As NHPs are closer to humans than rodents with respect to e.g. physiology, cognition, genetics, and immunology[3–10], results from NHP studies investigating cognition are likely more representative for the situation in humans.

In visual neuroscience, attention, object formation, categorization, and other aspects of cognition are extensively studied. In auditory neuroscience, several studies have also used different tasks (e.g. 2-alternative forced choice, go-no go) to probe different cognitive functions (such as memory, categorization, reward processing[11–14]). In general, though, studies in auditory cognition are lagging behind those of visual cognition with respect to overall sophistication of methods, experiments and task complexities. One factor for this is the common observation that monkeys have been notoriously difficult to train in the auditory domain, and generally display a bias towards vision. For example, it has been shown that baboons can easily learn to locate food items based on visual but not auditory cues[15]. Among other results, this surprising failure at such a seemingly simple auditory task has led the authors to suggest that inferential reasoning might be modality specific.

However, investigations into auditory capabilities and cognition increase in scope as NHPs have become genetically tractable organisms[1,2,16–18]. Notably, the common marmoset (*Callithrix jacchus*) has become a valuable model for biomedical research in general and the neurosciences in particular[19–21]. Factors such as the relative ease of breeding, early sexual maturation and short life span[22,23] have contributed to the rapid generation of genetic models of human mental and neurological diseases in marmosets[1,24–26]. While generally marmoset training is lacking behind the sophistication of cognitive NHP experiments traditionally performed with macaques, auditory capabilities of marmosets have been investigated extensively[27–32]. Furthermore, marmosets have now also become the go-to NHP model for hearing loss and cochlear implant research[33–36]. In the near future many more transgenic primate models will be developed which requires extensive phenotyping, as is standard for rodent models[37]. Phenotyping will need to investigate large number of subjects in a standardized and experimenter/observer-independent manner[38–44]. In addition, increased awareness for species-specific ethical demands asks for refinement of experimentation techniques as much as possible[45,46]. This has led to efforts developing home-cage, computer-based cognitive training of NHPs focusing on the visual domain[47–63].

To achieve comparable efforts in the auditory domain, there is a need for automatic, unsupervised cage-based training and testing of auditory tasks. Towards this goal, we built a standalone wireless device for auditory training and testing of common marmosets, directly in their own cage. The system, termed marmoset experimental behavioral instrument (MXBI), is mostly comprised of off-the-shelf or 3d printed components, is entirely programmed in Python, and based on the Raspberry Pi platform, for maximum flexibility of use, openness, and to allow for easy adaptation by others. The MXBI is set up with a server/client configuration in mind; and capable of animal tagging by means of radio-frequency identification (as in rodent systems[64]), which ultimately allows scalable, standardized, automated, and unsupervised training and testing protocols (AUT in short, from ref. [48]) in socially housed animals. Moreover, the MXBI and the procedures we describe contribute to the efforts of refining cognitive and environmental enrichments of NHPs in human care. Further, we report results from a set of four experiments: (1) an algorithm-based procedure for gradually and autonomously training naïve animals to the basics of a 2-Alternative-Choice task (2AC visual task); (2) an Audio-visual association experiment

where a conspecific call is contrasted to an artificial acoustic stimulus; (3) a generalization experiment assessing the flexibility of the acquired discrimination behavior with other stimuli; (4) and a psychoacoustic detection experiment for quantifying hearing thresholds in a cage-based setting. We show that marmosets can be trained to flexibly perform psychoacoustic experiments on a cage-based touchscreen device, via an automated and unsupervised training procedure that requires no human supervision and does not rely on fluid or food control, nor social separation.

## Results
In this study 14 adult common marmosets (*Callithrix jacchus*) of either sex and housed in pairs participated across one *initial training* phase and four autonomous cage-based experiments. Animals were generally trained in pairs on auditory tasks with a single MXBI attached to the animals' home cage and without fluid or social restrictions (Fig. 1a). Aside from the *initial training* (see below) all sessions ran autonomously, while an RFID module identified the animals and an algorithm controlled the individualized, performance-based progression in difficulty (see methods: Automated unsupervised training (AUT)).

**Initial training**. The goal of the *initial training* was to instruct naïve animals to interact with the touchscreen to receive liquid reward (Arabic gum or marshmallow solution) from the device's mouthpiece. The training was divided into three sequential steps: first, habituation to the device (supplementary video 1); second, forming a mouthpiece-reward association (supplementary video 2), and finally, a touch-to-drink phase (supplementary video 3.1 and 3.2). All animals started exploring the device from the very first session. During the touch-to-drink phase, a mesh tunnel was introduced inside the device (Fig. 1a), to allow only one animal at a time inside the MXBI. Animals were encouraged to enter the tunnel and reach the touchscreen by placing small pieces of marshmallows or arabic gum along the tunnel, on the mouthpiece, and on the screen. After the *initial training* was concluded (mean = 6 ± 1.4 sessions, Table 1), animals were introduced to the automated procedure gradually bringing them from naïve to experienced in discrimination as well as detection-based psychoacoustic tasks.

**General engagement on the MXBI across all autonomous experiments**. Individual animals engaged with the MXBI in different amounts with the median number of trials per session varying between 31 and 223. On average 116 trials per session (IQR = Q3-Q1 = 192) were performed (Fig. 1b, Table 1). While half of the animals had less than 10% of sessions without a single trial (median = 10.7%, IQR = 16.8%) two animals displayed more than 30% of sessions without performing a trial. On average 100 sessions were conducted per animal and 14 of those sessions had 0 trials (Fig. 1b). Controlling for session duration, we found no significant correlation between the total number of trials performed by each animal and session number (Partial Pearson correlation controlling for session duration; adjusted $r^2 = 0.05$, *p*-value: 0.1, $N = 802$; CI = −0.01, 0.13, Fig. 1c), suggesting that the level of engagement remained consistent across sessions. Qualitatively, animals tended to engage consistently throughout a session as indicated by the distribution of trial onset times (Fig. 1e). Consequently, the median time point at which half of the trials were performed was 0.52 of the session's duration (Fig. 1d).

**Automated unsupervised training (AUT)**. An automated and unsupervised training protocol (AUT[48]) was implemented to

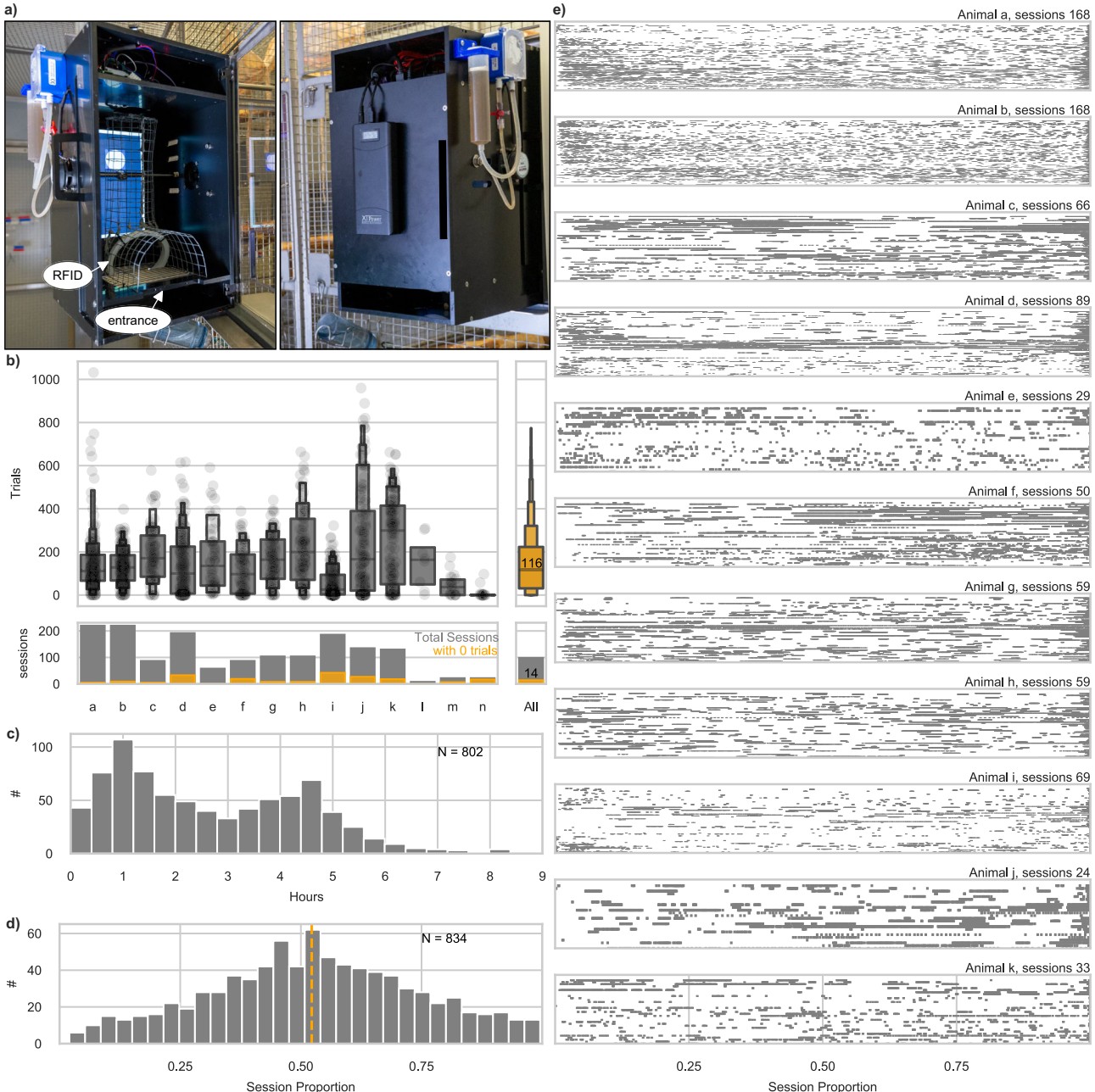

**Fig. 1 General engagement on the MXBI across all autonomous experiments. a** MXBI device attached to a cage in the animal facility. Left—opened for experimenter access to the inside of the device, Right—closed. **b** Letter-value plots of the number of trials performed in each session, dark gray: for each animal individually, orange: average distribution with all animals considered. The central box defines the median and 25th up to 75th percentile. Successively narrower boxes are drawn between the 1/8*100th and 7/8*100th, the 1/16th and 15/16*100th, and so on percentile. The total number of trials per animal can be found in Table 1. The number of sessions per animal and the average across animals are plotted below. Sessions without a single trial performed are given in orange. **c** Distribution of the duration of all sessions. **d** Distribution of all median timestamps as a function of session proportion. The dashed orange line indicates that across sessions half of the trials were performed within 52% of the session duration. **e** For each of the animals with more than 3000 trials, each trial of each session with more than 10 interactions (shared ordinated) is plotted with respect to its timestamp normalized by the session duration.

train naive marmosets at their own pace on the basics of a 2AC visually guided task. In order to identify the appropriate parameters upon which to build such autonomous procedure we first designed and tested multiple AUT versions with a subset of 9 animals (described in supplementary tables S1 and S2). The resulting final versions of the protocols (AUTs 8, 9, and 10), were then tested with 4 naïve animals (animals f, k, c, and d). The AUT procedure was comprised of 4 milestones—(1) decrease of the size of a visual stimulus (trigger) to be touched for reward, (2)

change of position of a visual stimulus, (3) introduction of sound and delayed presentation of a visual target, (4) introduction of a second visual target as a distractor—that unfolded through a total of 48 dynamic steps (Fig. 2; Fig. S4C). During each session the transitions between steps and milestones were based on the animal's performance in a sliding window of 10 trials (hit rate of > 80% to advance, <= 20% to retreat; Fig. S4D). Figure 2c shows the hit rate across individual steps and milestones for the 4 naïve animals that only performed the final versions of the AUT. While

**Table 1 Characteristics and statistics of all animals involved in the experiments.**

| ID | Cage mate ID | Characteristics | | | Initial training (Sessions) | | | Trials (across all tasks) | Sessions (across all tasks) | Sessions with 0 trials |
|---|---|---|---|---|---|---|---|---|---|---|
| | | Sex | Age [months] | Weight [g] | Habituation | Mouthpiece association | Touch-to-drink | | | |
| a | b | f | 41 | – | – | – | 3 | 31081 | 220 | 5 |
| b | a | m | 36 | – | – | – | 4 | 28222 | 220 | 9 |
| c | f | m | 24 | 415 | 1 | 2 | 3 | 16181 | 87 | 5 |
| d | i | f | 84 | 375 | 1 | 2 | 2 | 25941 | 192 | 33 |
| e | l | f | 26 | 423 | 2 | 1 | 3 | 9822 | 58 | 0 |
| f | c | m | 84 | 386 | 1 | 2 | 2 | 9901 | 87 | 19 |
| g | h | m | 29 | 476 | 2 | 3 | 2 | 17296 | 104 | 8 |
| h | g | m | 32 | 354 | 2 | 2 | 3 | 23157 | 104 | 8 |
| i | d | m | 26 | 390 | 0.5 | 0.5 | 1 | 10265 | 186 | 42 |
| j | k | m | 33 | 446 | 3 | 2 | 2 | 32585 | 135 | 27 |
| k | J | f | 31 | 388 | 2 | 2 | 2 | 33424 | 130 | 18 |
| l | e | f | 32 | 471 | 2 | 2 | 3 | 1212 | 8 | 0 |
| m | n | m | 45 | – | – | – | – | 975 | 22 | 7 |
| n | m | f | 31 | 366 | 2 | 2 | 3 | 168 | 22 | 17 |

"Characteristics" columns report the sex (Sex), age in months (Age [months]), and weight in grams (W [g]) of each animal at the start of experiment. Initial training columns report the number of sessions required for the shaping stages habituation (Habituation), mouthpiece-reward association (Mouthpiece association) and "touch-to-drink" (Touch-to-drink). Columns: Trials, Sessions and Session with 0 trials report the statistics for the corresponding panel in Fig. 1 regarding the total number of trials ("Trials (across all tasks)") and the total number of sessions collected for each animal ("Sessions (across all tasks)"). The column "Sessions with 0 trials" summarizes the amount of sessions without interactions. The Initial Training.

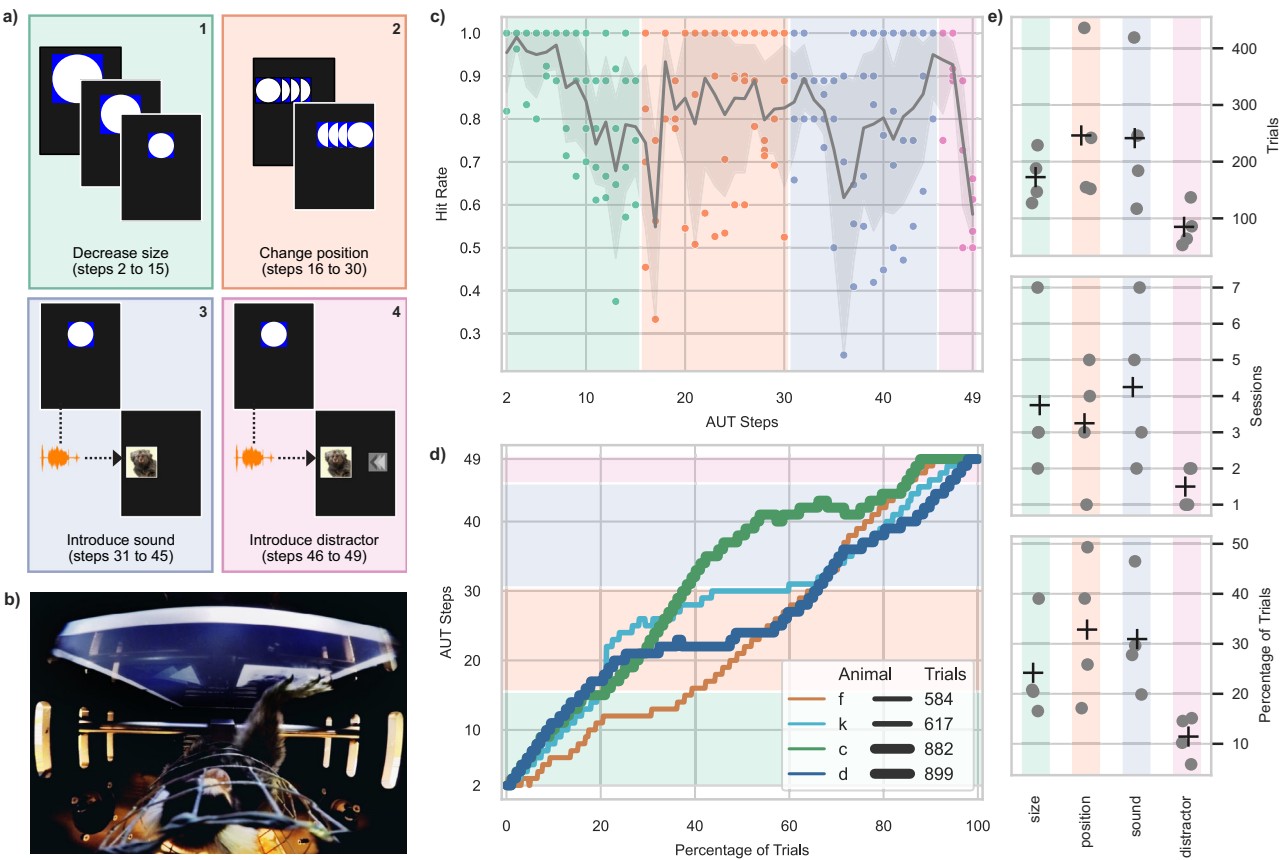

**Fig. 2 Automated unsupervised training (AUT) performance across four representative animals. a** Schematics of the four main milestones of the final AUT protocol. **b** Picture taken with an external high-resolution wide-angle camera, where an animal can be seen completing a trial. **c** Average hit rate as a function of steps (gray area represents the 95% confidence interval of the mean across animals) across the four animals considered in this analysis. **d** Percentage of trials spent on each AUT step and milestones (shaded background) with line thickness indicating the total amount of trials performed by the corresponding animal. **e** Distributions of number of trials, number of sessions, and percentage of total trials for each milestone across the four naïve animals (crosses represent average values).

the procedure was designed to encourage a smooth transition from step to step, certain steps (and thus milestones) required more trials to be accomplished. As a consequence, the hit rate calculated across animals varies substantially as a function of AUT step (Fig. 2c). Due to animals learning at different paces and performing different number of trials, we quantified the progression through the AUT as a function of the percentage of total trials completed by each animal (Fig. 2d). This allowed us to visualize and compare learning progress across animals with inherently different working paces on a common frame of reference. Both the total amount of trials (expressed by line thickness in Fig. 2d) needed to complete the AUT and the learning curves throughout the AUT vary substantially across animals (Fig. 2e) in the middle portion of the AUT, during which the stimulus changed position on the screen and an acoustic stimulus was introduced. Starting from the introduction of sound (milestone 3) we introduced timeouts (gray screen) to provide further feedback on wrong trials. Analysis of inter-trial-intervals (ITIs) trials revealed shorter average ITIs after correct vs. wrong trials suggesting an effect of timeouts on animal behavior (Fig. S3 and Table S4).

**Audio-visual association**. Next, we tested whether animals would generalize from the visually guided 2AC task introduced via the AUT procedure to an acoustically guided 2AC discrimination. In this experiment animals were required to discriminate between a conspecific juvenile call (in the following referred to as voc), and a pure tone (simple train—sTr—chosen for individual animals from a range between 1.5 and 3.5 kHz), by selecting one of two visual stimuli permanently associated with each sound (supplementary Video 4). 5 out of 9 animals successfully learned to discriminate between the sTr and the voc by selecting a geometric pattern or a conspecifics face, respectively (Fig. 3a, c). The remaining 4 animals performed at chance level. To disentangle if these animals were unable to solve the task or maybe were unwilling to perform above chance, we devised a 3 alternative-choice (3AC; upon sound presentation animals had to choose between 3 visual symbols, see methods) version of the same task (Fig. 3b, c) and tested 2 of these animals and 2 additional animals who had failed a different control condition (see supplementary material: Artificial Discrimination, Figs. S1, S2). In the 3AC task, all 4 animals performed the task significantly above chance (Binomial test, pot-hoc corrected for multiple comparisons; Table 2). Taken together these results demonstrate that 9 out of 11 animals learned the Audio-visual association. The remaining two animals that did not learn the 2AC discrimination were assigned to a different project and were not tested on the 3AC version. Additionally, 7 out of 9 animals who accomplished the discrimination task exhibited significantly longer reaction times in responding to the target in voc vs sTr trials (Fig. 3d; Table 2), indicating that the animals behaved differently for different acoustic stimuli.

**Generalization to novel stimuli**. With the five best performing animals in the audio-visual association experiment, we assessed whether animals would be able to generalize the acquired discrimination to three novel stimuli (Fig. 4a): two different types of vocalizations—an adult marmosets' Phee and a Twitter—and a white noise sound. All 5 animals quickly learned to discriminate the Twitter and the Phee when contrasted to the sTr (Fig. 4b, c). On the other hand, when two new stimuli were contrasted with each other animals displayed lower hit rates. In the white noise vs Twitter condition, 3 animals acquired the discrimination; 1 animal displayed a bias towards the twitter it had previously learned; and for 1 animal the performance fluctuated between 0.6 and 0.75

in the sessions prior to the last 2 in which it was not significantly different from chance. When the juvenile vocalization (voc) was juxtaposed to the Twitter only 2 animals significantly performed above chance and another performed significantly above chance only for the Twitter. Animals seemed to find it more difficult discriminating between vocalizations, despite having already learned and successfully discriminated both from other stimuli extensively (see Table 3). We interpret this result as an indication that vocalization stimuli (voc, Twitter, and Phee) carry a distinctive meaning to the animals compared to more artificial stimuli (tones or white noise). This could in fact explain why animals readily discriminate them when contrasted to artificial stimuli but do not display significant discrimination between multiple vocalizations. Note that Animal $i$ was not quantified in the voc vs Twitter and in the Phee vs sTr condition due to a limited number of trials (less than 50 trials in each task).

**Psychoacoustic assessment of stimulus thresholds**. Last, we addressed whether the MXBI can be employed for psychoacoustics. We chose to investigate hearing thresholds in a vocalization-detection task and towards this goal trained three animals (animal a, b, and d). In this experiment animals that already knew the association between the acoustic and corresponding visual stimuli (see above: section "audio-visual association"), were now trained to associate the absence of the vocalization with the visual stimulus for the sTr (Fig. 5). The method of constant stimuli was employed by randomly selecting the sound level from a set of values between 0 and 80 dB SPL. The animals were required to report the presence or absence of the vocalization by touching the marmoset face (visual stimulus coupled with the voc) or the triangles (visual stimulus coupled with silence), respectively. Note that due to the nature of the task, reward to the animals for stimuli in the range between 15- and 45-dB SPL was provided regardless of the animal's choice. This was instrumental to prevent frustration and thus disengagement from the task when the acoustic stimulus was presented at amplitudes presumably close to the animal's hearing thresholds. In contrast, reward was dependent on the animals' choice for stimuli at and above 60 dB SPL and at 0 dB SPL. The aim of this reward scheme, illustrated in Fig. 5a, was to encourage the animals to use the triangles and the marmoset face as yes/no options for the presence/absence of the acoustic stimulation. After two to three sessions with only high amplitude stimuli (70 dB SPL) to stabilize the animals' discrimination performance at 75% or above, test sessions commenced (three for animal d and four sessions for animals a and b—Fig. 5b). The estimated hearing threshold for the vocalization stimulus (mean 37.3 dB SPL; 36 for animal a, 49 for animal b, 27 for animal d) was below the background noise of the facility of 60 dB SPL (measured inside the MXBI with a measurement microphone and amplifier, see methods; spectrograms of 3 representative 1 min long recordings are shown in Fig. 5c).

## Discussion

In this study, we report results from four sequential experiments conducted with a stand-alone, touchscreen-based system—termed MXBI—tailored to perform training as well as psychophysical testing of common marmosets in auditory tasks. Animals involved in this experiment operated the device with a consistent level of engagement and for a prolonged time, directly in their own housing environment, without dietary restriction or social separation. All animals navigated an automated, unsupervised training procedure with ease and at their own pace, going from naïve to experienced in a visually guided discrimination task. In a following audio-visual association experiment, nine out of eleven

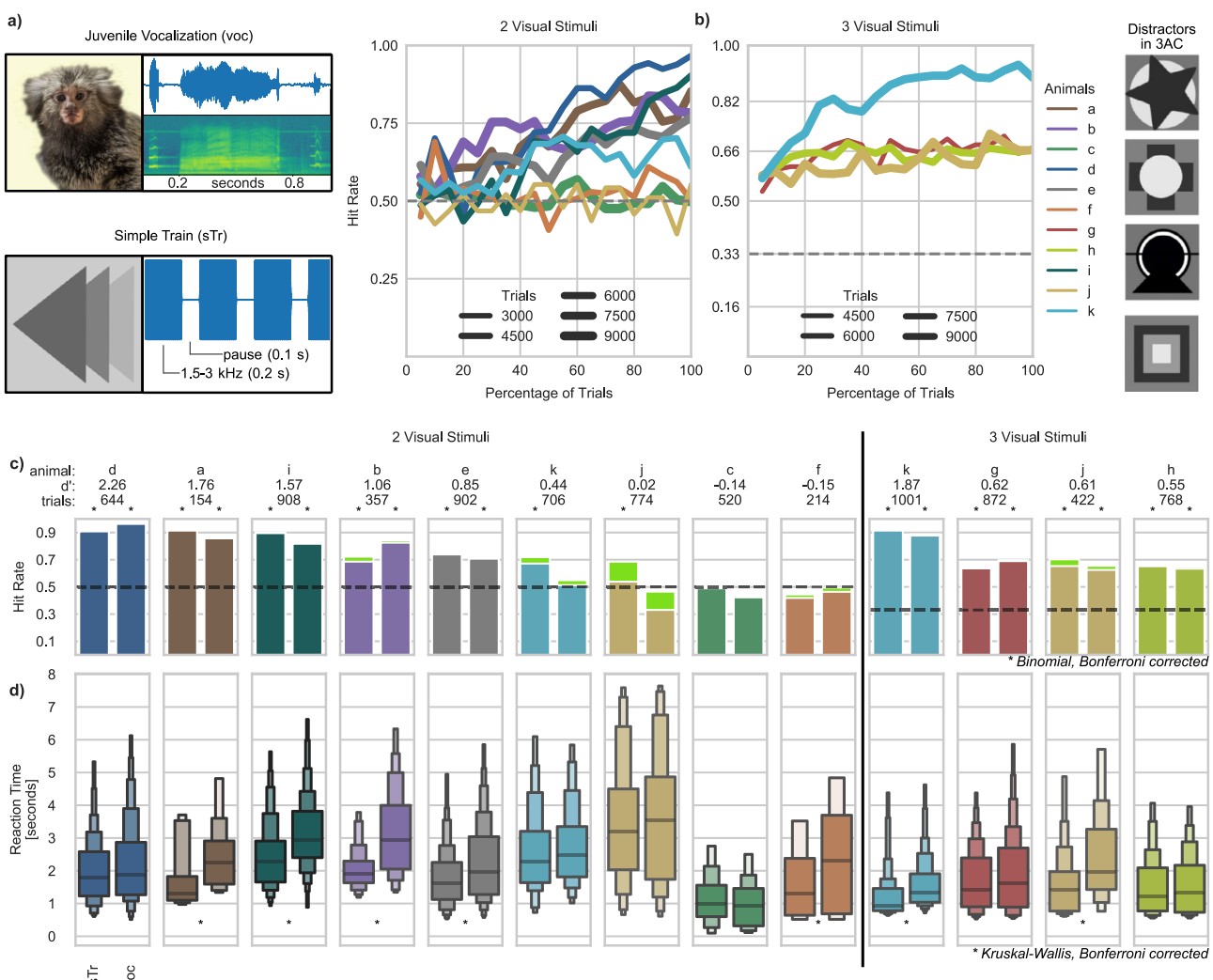

**Fig. 3 Stimuli and results from the Audio-visual association experiment. a, b** Visual and acoustic stimuli combinations used and hit rate as a function of percentage of trials performed, for different animals (colored lines) and across tasks. Hit rate, as a function of the percentage of trials performed by each animal, is grouped into bins of 5%. Line thickness represents the number of trials of each animal in each panel. Dashed lines at 0.5 and 0.33 represent the chance level for the two tasks. **c** Hit rate across the last 5 sessions as a function of stimulus type ("sTr" for the pure tone stimulus, "voc." for the juvenile vocalization; green bars indicate ignored trials), with corresponding number of trials and sensitivity index (d'). Stars represent significance reached for the given stimulus at a Bonferroni post-hoc corrected Binomial test (one-sided test). **d** Letter-value plots of the reaction times plotted for each stimulus type separately. The central box defines the median and 25th up to 75th percentile. Successively narrower boxes are drawn between the 1/8*100th and 7/8*100th, the 1/16*100th and 15/16*100th, and so on, percentile. Stars represent significant statistical difference in reaction times between the two stimuli at a Bonferroni post-hoc corrected Kruskal–Wallis Test (one-sided test). Statistics and N number for panels (**c**) and (**d**) are given in Table 2.

animals further acquired proficiency in an acoustically guided 2AC or 3AC discrimination task. Animals also quickly learned to flexibly discriminate three novel sounds they had never encountered before in a generalization experiment. Finally, we assessed the hearing thresholds of 3 animals with a spectro-temporally complex sound under potentially distracting auditory conditions. Our results indicate that: (1) marmoset monkeys consistently engage in various psychoacoustic experiments; (2) while performing enough trials and at high performance to allow psychometric evaluations; (3) in a self-paced manner; (4) without the need of dietary restriction or separation from their peers; and (5) with high degree of training flexibility.

**Home-cage training of naïve animals.** For our experiments we designed a cage-based device and employed an unsupervised algorithm to gradually and autonomously make naïve marmosets accustomed to a 2 or 3 alternative-choice task and a simple

detection task in the auditory modality. Each of the 14 animals who participated and successfully completed the first experiment learned (1) to seek and consume reward delivered from the mouthpiece; (2) to operate a touchscreen proficiently; (3) to respond with appropriate timing to abstract sensory stimulation; (4) to understand the concept of a trial structure; (5) to tolerate frustration when failing a trial; 6) and ultimately to continuously devise, update, and deploy problem-solving strategies. For practical, experimental, as well as ethical reasons, we aimed at developing an experimental protocol to train many of these aspects directly in the animals' own housing environment, at the animals' own pace[47,51,53,56,65,66], and without dietary restrictions. Most of these aspects were instructed by a computerized training strategy in which the difficulty was automatically adjusted according to the trial-to-trial performance of the individual animal. The Automated Unsupervised Training (AUT) consisted of a pre-programmed series of steps in which several elements of the task were slowly introduced or adjusted, from trial to trial. The

**Table 2 Summary statistics for the audio-visual association across animal and stimuli (Fig. 3c, d).**

| Animal | Task | Stimulus | Hitrate | Trials | d' | Binomial test on performance (Fig. 3c) | | | Kruskal-Wallis test on Reaction Times (Fig. 3d) | | | | | |
|---|---|---|---|---|---|---|---|---|---|---|---|---|---|---|
| | | | | | | N (w/o ignored) | Degrees of freedom | Binomial Test (adjusted p-value) | Median | IQR | N | Degrees of freedom | Test Statistics | Kruskal-Wallis (adjusted p-value) |
| a | 2AC | sTr | 0.91 | 70 | 1.76 | 70 | 1 | **3.18E−12** | 1.3065 | 0.72725 | 64 | 1 | 28.22 | **2.81E−06** |
| | | voc | 0.86 | 84 | | 83 | 1 | **5.09E−11** | 2.2635 | 1.3195 | 72 | 1 | | |
| b | 2AC | sTr | 0.69 | 191 | 1.06 | 184 | 1 | **1.08E−07** | 1.904 | 0.6625 | 131 | 1 | 54.65 | **3.74E−12** |
| | | voc | 0.83 | 166 | | 163 | 1 | **2.71E−18** | 2.95 | 1.944 | 137 | 1 | | |
| c | 2AC | sTr | 0.49 | 241 | -0.14 | 238 | 1 | 1 | 0.999 | 0.95925 | 118 | 1 | 1.52 | 1 |
| | | voc | 0.42 | 279 | | 278 | 1 | 1 | 0.928 | 1.14275 | 118 | 1 | | |
| d | 2AC | sTr | 0.91 | 316 | 2.26 | 316 | 1 | **2.01E−53** | 1.797 | 1.3515 | 287 | 1 | 2.73 | 1 |
| | | voc | 0.96 | 328 | | 326 | 1 | **6.38E−79** | 1.864 | 1.60225 | 316 | 1 | | |
| e | 2AC | sTr | 0.74 | 444 | 0.85 | 444 | 1 | **2.15E−23** | 1.634 | 1.132 | 328 | 1 | 21.99 | **7.14E−05** |
| | | voc | 0.71 | 458 | | 453 | 1 | **2.59E−19** | 1.9605 | 1.76475 | 324 | 1 | | |
| f | 2AC | sTr | 0.42 | 117 | -0.15 | 114 | 1 | 1 | 1.308 | 1.737 | 49 | 1 | 3.84 | 1 |
| | | voc | 0.46 | 97 | | 94 | 1 | 1 | 2.308 | 3.01 | 45 | 1 | | |
| g | 3AC | sTr | 0.64 | 437 | 0.62 | 432 | 1 | **3.32E−08** | 1.4295 | 1.493 | 278 | 1 | 2.20 | 1 |
| | | voc | 0.69 | 435 | | 432 | 1 | **5.18E−15** | 1.626 | 1.805 | 300 | 1 | | |
| h | 3AC | sTr | 0.65 | 388 | 0.55 | 384 | 1 | **6.21E−09** | 1.216 | 1.318 | 253 | 1 | 0.27 | 1 |
| | | voc | 0.63 | 380 | | 375 | 1 | **4.66E−07** | 1.319 | 1.436 | 241 | 1 | | |
| i | 2AC | sTr | 0.9 | 459 | 1.57 | 457 | 1 | **3.11E−73** | 2.289 | 1.2525 | 411 | 1 | 83.00 | **2.14E−18** |
| | | voc | 0.82 | 449 | | 444 | 1 | **3.19E−45** | 2.946 | 1.4155 | 367 | 1 | | |
| j | 2AC | sTr | 0.54 | 299 | 0.02 | 255 | 1 | **0.0004** | 3.191 | 2.469 | 161 | 1 | 0.03 | 1 |
| | | voc | 0.33 | 475 | | 411 | 1 | 1 | 3.554 | 3.128 | 157 | 1 | | |
| j | 3AC | sTr | 0.65 | 222 | 0.61 | 211 | 1 | **7.25E−07** | 1.433 | 1.204 | 145 | 1 | 26.39 | **7.26E−06** |
| | | voc | 0.63 | 200 | | 194 | 1 | **0.0009** | 1.972 | 1.841 | 125 | 1 | | |
| k | 2AC | sTr | 0.67 | 320 | 0.44 | 305 | 1 | **7.70E−12** | 2.282 | 1.567 | 215 | 1 | 2.64 | 1 |
| | | voc | 0.51 | 386 | | 372 | 1 | 1 | 2.467 | 1.53675 | 198 | 1 | | |
| k | 3AC | sTr | 0.91 | 503 | 1.87 | 498 | 1 | **4.86E−92** | 0.919 | 0.67875 | 460 | 1 | 120.62 | **1.20E−26** |
| | | voc | 0.88 | 498 | | 493 | 1 | **4.01E−73** | 1.337 | 0.869 | 437 | 1 | | |

"Binomial test on performance (Fig. 3c)" report information regarding the statistical deviations of performance (across stimuli and task type) from a theoretically expected distribution of observations (one-sided), with p-values adjusted with a post-hoc Bonferroni correction for multiple comparisons. D-prime value is provided as indication of the sensitivity of each animal on given task. Columns under "Kruskal-Wallis test on Reaction Times (Fig. 3d)" report information regarding the statistical difference of the reaction time to the sTr and the voc stimuli, with p-values adjusted with a post-hoc Bonferroni correction for multiple comparisons. Significant values are indicated in bold font.

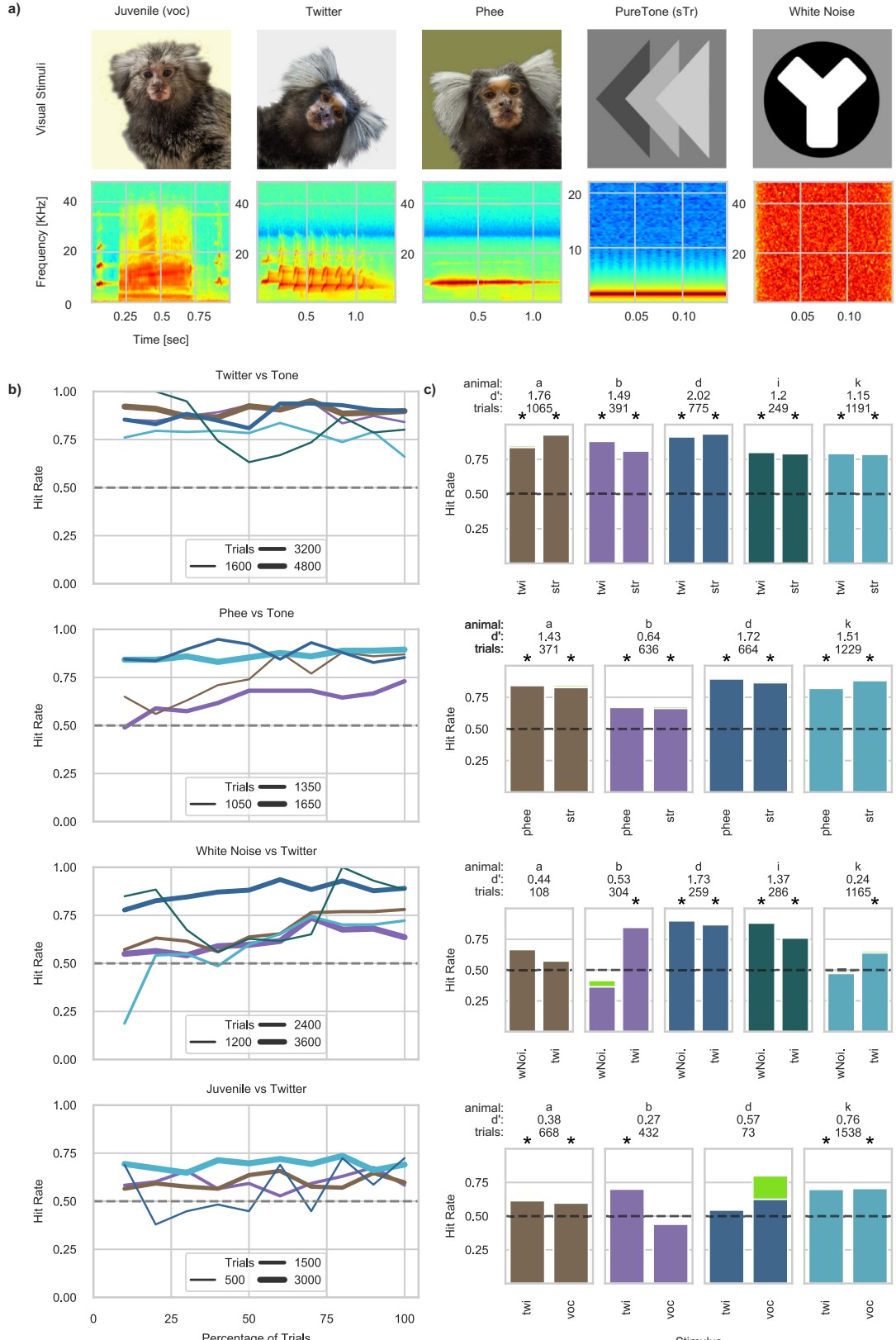

aim of this strategy is to keep animals at a comfortable level of performance to presumably limit frustration, while making the task gradually more difficult and thus making the animals more and more proficient[48]. Additionally, such subtle, gradual, and constant change in the challenge offered to the animals has been suggested to prevent loss of interest[67–70]. We indeed observed a long-term rate of engagement, across several hundred sessions across all animals, suggesting an interest in the experimental sessions that could not be attributed solely to novelty[69]. Additionally, animals were generally kept together with their cage mate in their home-enclosure and were fed normal colony diet, prior to, after or even during the sessions. Fluid was also available

**Fig. 4 Generalization of audio-visual association to novel stimuli. a** Representative visual stimuli and spectrograms for all five stimuli used in the experiment, paired column wise. The juvenile vocalization (voc) and the pure tone (simple train—sTr) are the same stimuli used in the previous experiments (Fig. 2). **b** Hit rate as a function of percentage of trials (10% bin) across of the five animals and the four tasks, with line thickness representing the total amount of trials of each animal at each task. **c** Hit rate as a function of stimulus in the last three sessions (eight sessions for animal d in the condition Juvenile vs Twitter and nine for animal k in White Noise vs Twitter), with corresponding number of trials and sensitivity index (d'). Star represents significance reached at a Bonferroni post-hoc corrected Binomial test for the corresponding stimulus (one-sided test). Dashed lines across all plots represent the 50% chance threshold. Green indicates ignored trials. The performance of animal d in the task Juvenile vs Twitter and of animal i in the task Twitter was Tone were based on eight sessions and nine sessions, respectively, instead of 3 (like the rest of the animals and tasks). This was necessary to consider a number of total trials higher than 40 and thus increase the statistical reliability of testing the performance of each animal against chance.

**Table 3 Summary statistics for the Generalization to novel stimuli across animals and the four conditions (Fig. 4).**

| Animal | Task | Sound | Trials | Hit Rate | Binomial test on performance (Fig. 4c) | | |
|---|---|---|---|---|---|---|---|
| | | | | | N | Degrees of freedom | Adjusted p-value |
| a | Twitter vs Tone | Tone | 530 | 0.93 | 529 | 1 | **3.75E−100** |
| | Phee vs Tone | Tone | 186 | 0.83 | 183 | 1 | **1.78E−20** |
| | Juvenile vs Twitter | Juvenile | 333 | 0.6 | 333 | 1 | **8.72E−03** |
| | Phee vs Tone | Phee | 185 | 0.84 | 184 | 1 | **1.94E−21** |
| | Twitter vs Tone | Twitter | 535 | 0.84 | 528 | 1 | **5.05E−61** |
| | Juvenile vs Twitter | Twitter | 335 | 0.61 | 334 | 1 | **4.65E−04** |
| | Noise vs Twitter | Twitter | 54 | 0.57 | 54 | 1 | 1 |
| | Noise vs Twitter | Noise | 54 | 0.67 | 54 | 1 | **3.97E−01** |
| b | Twitter vs Tone | Tone | 200 | 0.81 | 198 | 1 | **5.52E−19** |
| | Phee vs Tone | Tone | 320 | 0.66 | 315 | 1 | **1.59E−08** |
| | Juvenile vs Twitter | Juvenile | 218 | 0.44 | 217 | 1 | 1 |
| | Phee vs Tone | Phee | 316 | 0.67 | 314 | 1 | 1 |
| | Twitter vs Tone | Twitter | 191 | 0.88 | 190 | 1 | **9.98E−28** |
| | Juvenile vs Twitter | Twitter | 214 | 0.7 | 214 | 1 | **7.76E−08** |
| | Noise vs Twitter | Twitter | 150 | 0.85 | 149 | 1 | **7.60E−18** |
| | Noise vs Twitter | Noise | 154 | 0.36 | 146 | 1 | 1 |
| d | Twitter vs Tone | Tone | 389 | 0.93 | 389 | 1 | **7.86E−76** |
| | Phee vs Tone | Tone | 328 | 0.87 | 324 | 1 | **3.57E−45** |
| | Juvenile vs Twitter | Juvenile | 40 | 0.63 | 33 | 1 | **9.10E−02** |
| | Phee vs Tone | Phee | 336 | 0.9 | 335 | 1 | **2.74E−53** |
| | Twitter vs Tone | Twitter | 386 | 0.91 | 386 | 1 | **1.87E−66** |
| | Juvenile vs Twitter | Twitter | 33 | 0.55 | 33 | 1 | 1 |
| | Noise vs Twitter | Twitter | 129 | 0.87 | 129 | 1 | **4.89E−17** |
| | Noise vs Twitter | Noise | 130 | 0.9 | 129 | 1 | **1.71E−21** |
| i | Twitter vs Tone | Tone | 129 | 0.79 | 128 | 1 | **1.56E−10** |
| | Phee vs Tone* | Tone | 9 | 0.89 | 9 | 1 | **7.81E−01** |
| | Juvenile vs Twitter* | Juvenile | 1 | 1 | 1 | 1 | 1 |
| | Phee vs Tone* | Phee | 10 | 0.7 | 10 | 1 | 1 |
| | Twitter vs Tone | Twitter | 120 | 0.8 | 119 | 1 | **1.71E−10** |
| | Juvenile vs Twitter* | Twitter | 4 | 0.5 | 4 | 1 | 1 |
| | Noise vs Twitter | Twitter | 142 | 0.76 | 141 | 1 | **3.37E−09** |
| | Noise vs Twitter | Noise | 144 | 0.88 | 143 | 1 | **2.50E−21** |
| k | Twitter vs Tone | Tone | 590 | 0.79 | 588 | 1 | **7.60E−46** |
| | Phee vs Tone | Tone | 597 | 0.88 | 596 | 1 | **5.35E−87** |
| | Juvenile vs Twitter | Juvenile | 751 | 0.71 | 749 | 1 | **2.69E−29** |
| | Phee vs Tone | Phee | 632 | 0.82 | 629 | 1 | **4.03E−63** |
| | Twitter vs Tone | Twitter | 601 | 0.79 | 598 | 1 | **5.14E−49** |
| | Juvenile vs Twitter | Twitter | 787 | 0.7 | 785 | 1 | **1.25E−28** |
| | Noise vs Twitter | Twitter | 541 | 0.64 | 532 | 1 | **7.54E−11** |
| | Noise vs Twitter | Noise | 624 | 0.47 | 615 | 1 | 1 |

Columns under " Binomial test on performance (Fig. 4c)" report information regarding the statistical deviations of performance (across stimuli and task type) from a theoretically expected distribution of observations (one-sided), with p-values adjusted with a post-hoc Bonferroni correction for multiple comparisons. Significant values are indicated in bold font. Stars behind the comparison in the column "Task" indicate low trial numbers performed suggesting a low statistical power.

ad libitum. Such generalized and continued interest towards the MXBI, free of any additional coercion, was presumably the result of the combination of a highly preferred primary reinforcer (liquid arabic gum or marshmallow solution), a cognitive, sensory, and interactively rich environment[67,71,72], and the dynamical adjustments in task level[48,70]. Moreover, we did not observe any behavioral alteration that would suggest excessive attachment to our system at the level of the single individual or cage-pair of animals. Rather, 50% of the trials occurred within the first half of the session, in line with a recent report of a steady rate of interactions in voluntary training of motor tasks throughout the waking hours[66].

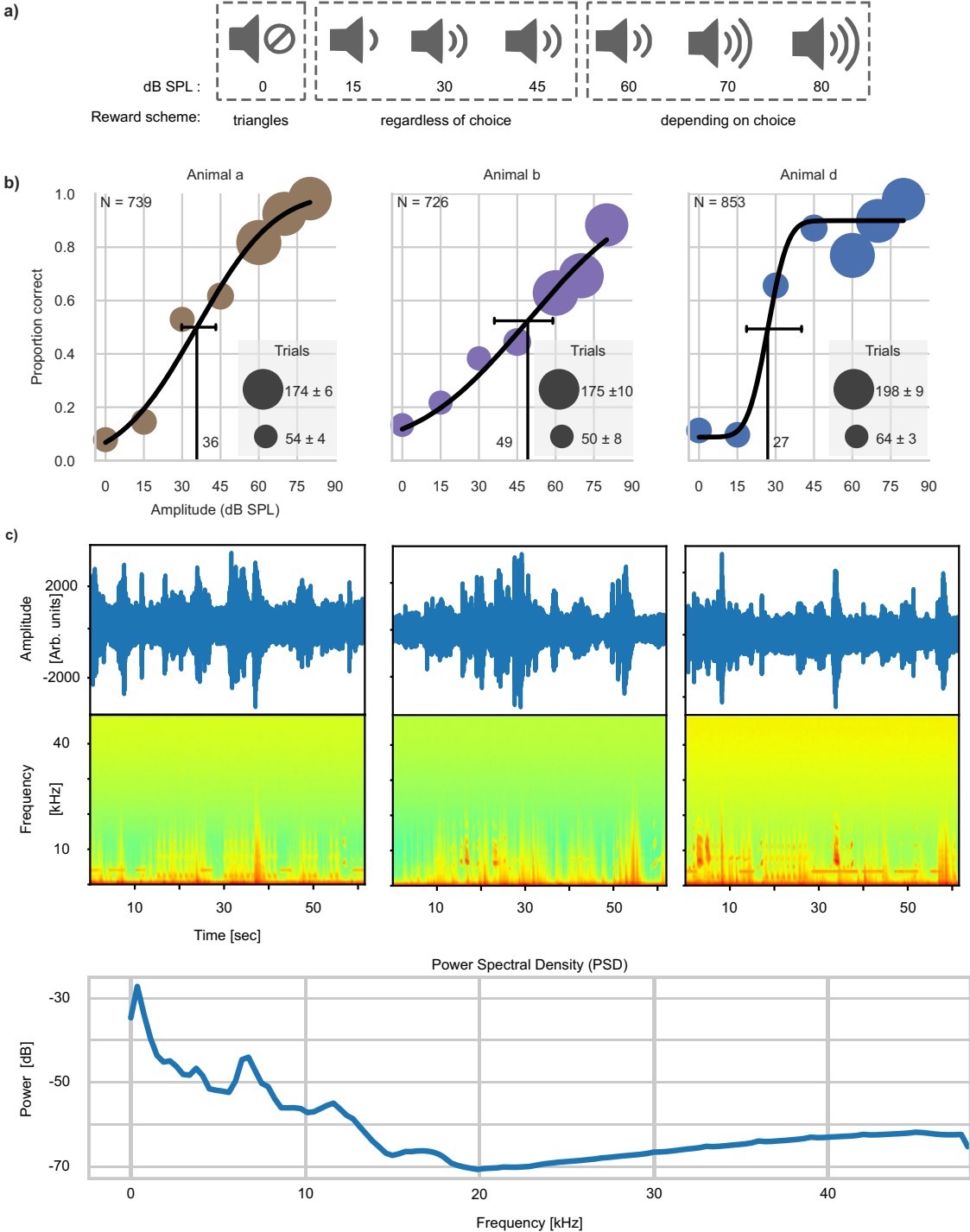

**Fig. 5 Psychophysical assessment of hearing thresholds for animals a, b, and d after training on the Audio-visual association experiment. a** Schematic representation of the sound levels used and the reward scheme associated with each level. Upon presentation of the vocalization stimulus: at 0 dB SPL, only the selection of the triangles was rewarded. For intensities 15, 30, and 45 dB SPL the animal was rewarded regardless of choice. For intensities at and above 60 dB SPL the reward was delivered depending on the animal's choice. **b** Psychometric estimation of hearing thresholds (black vertical lines) based on the proportion of times the animals selected the correct response across the intensities used. 95% confidence intervals (CI) are indicated with black horizontal lines (Animal a: threshold 36 dB SPL; CI between 29 and 43; b: 49 dB SPL, CI between 36 and 58; d: 27 dB SPL, CI between 27 and 40). **c** waveforms and spectrograms of 3 one-minute-long snippets recorded inside an MXBI while another MXBI in the same colony room was used to gather data for the audio-visual association experiment. Bottom panel shows the power spectral density of a 5 h long recording with dominant peaks at 6, 12, and 18 kHz caused by vocalizations.

Finally, because we instructed tasks that are typical in cognitive neuroscience and animal cognition (namely a two or three alternative-choice and a detection task), we believe that similar results would be achieved in training as well as testing other sensory or cognitive domains.

**Training flexibility of marmosets**. With the exception of two animals who were assigned to a different project and could not be trained further, all animals were successfully trained and tested in audio-visual association experiments reported here. It is important to note that while two animals—a and b—readily transferred the knowledge acquired in the visually guided discrimination (Automated Unsupervised Training) to quickly learn the acoustically guided discrimination (audio-visual association), the remaining seven animals required a substantial amount of trials to reach the same level of proficiency. Three animals out of the remaining seven also rapidly generalized the acquired discrimination to novel acoustic stimuli at a comparable rate to animals a and b. Therefore, while the initial transition from the visual to the acoustic domain occurred at variable speed, all tested animals showed a comparable level of flexibility in generalizing to novel stimuli. Finally, all three animals tested in the psychoacoustic assessment, quickly learned to reinterpret the discrimination as a detection task as soon as the reward scheme was adjusted. This allowed for a systematic psychoacoustic assessment of the sound intensity required to detect a vocalization under conditions with background noise.

Together, our results suggest a high degree of training flexibility of common marmosets in general and the auditory modality in particular. Specifically, marmosets can: (1) transfer acquired rules from the visual to the acoustic domain; (2) rapidly learn to discriminate novel acoustic stimuli and (3) flexibly reinterpret a discrimination task as a detection task.

**Cognitive hearing in marmosets**. The success of the acoustic experiments presented in this study could partly be due to intrinsic properties of the stimuli employed based on the naturalistic connotation in both the visual and the acoustic domain of the juvenile vocalization and juvenile marmoset face association. This 'natural association' might then also support the association of the respective other stimuli. Our failed attempts, detailed in the supplementary material, indeed demonstrate the difficulty in having marmosets associate artificial stimuli across the auditory and visual modality. The guiding strategy was that additional properties of the stimuli should match across modalities to support crossmodal association and considered successful concepts from training of rodents and ferrets[73,74]. For example, we presented auditory and visual stimuli together with a reward, or a timeout screen, in a temporally overlapping fashion which leads to strong associations of stimulus components in rodents. Also, the sound was presented from the speaker on which side the correct visual response indicator was located. This has been shown to be a strong cue for ferrets to guide choice towards the respective sound direction. In stark contrast, none of these approaches were successful in marmosets.

Results from the generalization experiment indicate that animals could quickly and flexibly learn to discriminate novel auditory stimuli. On the other hand, when two different types of vocalizations were contrasted, only two animals out of 4 performed above chance. Taken together these results indicate that (1) vocalizations might carry a distinctive meaning to the animals that can be exploited to train common marmosets on various psychoacoustic tasks; and (2) the use of a combination of naturalistic and artificial sounds is more likely to instruct marmosets in performing psychoacoustic tasks above chance level.

**Psychoacoustic assessment of marmosets in the home enclosure**. Performing auditory psychophysics directly in the animals' colony poses an acoustically challenging environment due to the uncontrolled background noise. The sound pressure needed in order to detect a vocalization of a juvenile marmoset in a cage-based setting—37.3 dB SPL—was below the sound level of the facility's background noise—~60 dB SPL. This might be explained by the adaptation of the auditory system to background sounds which has been documented along the auditory pathway[75–79] and has been suggested to optimize perception to the environment[76,77]. Additionally, the juvenile vocalization might have been less affected by background noise (mostly driven by ventilation and marmoset vocalizations) as it minimally overlaps the sound spectrum typically encountered in our colony of adult animals. Nonetheless, our data show that NHP's psychoacoustic training and assessment is feasible within the animals' home enclosure similar to chair based psychophysics[29]. While measurements of hearing thresholds in more classical controlled settings are essential to understand auditory processing and sensitivity, the investigation of audition in more naturalistic environments could provide a closer estimate of real-world hearing capabilities. This might be particularly relevant for auditory processes and mechanisms that involve higher-level, top-down, cortical influences[80–82] and thus are more susceptible to the influence of environmental contextual factors. Environmental sounds produced by conspecifics, for example, could affect how task-relevant sounds are encoded, processed, and interpreted by marmosets that heavily rely on acoustic communication to cooperate, live together, and survive[83].

**Towards a high-throughput pipeline for auditory neuroscience**. The development of transgenic primate models – and especially marmoset models—for various human diseases[1,24,84,85] will require phenotyping a large number of animals similar to mouse phenotyping pipelines[37,38,86]. Consequently, cognitive training and testing paradigms, designed around the marmoset model, need to be developed, tested, and implemented[70,87]. Furthermore, in order to allow high-throughput training and testing of common marmosets directly in their own housing environment, our device was designed and built with a series of hardware and software features in mind. First, the use of an inexpensive single-board computer as central control unit of the whole device allows for straightforward scaling to more devices and simple adaptation to new experimental requirements. To the best of our knowledge, besides the MXBI introduced here, a fully wireless cage-based system tailored towards visuo-acoustic stimulation and training, capable of ID tagging and set up to be server/client ready has not been presented yet. The wireless connectivity of the MXBI, allowed us to build a network of devices that autonomously interact with a single server node. Upon booting of an MXBI a series of scripts ensures that each device is connected to the central hub where (1) information about animals' ID are stored (used for matching ID codes coming from the implanted chips), (2) data are routinely backed up from the device, and (3) the videos of the sessions are stored. Besides having a unique network address, all devices are essentially identical and can therefore be used on any suitable home cage in our colony. Upon crossing the RFID coil, information coming from the implanted chip will be matched with the database on the server and the local device will load the desired task and AUT step for the given animal. Furthermore, employing a battery-based power solution for the MXBI made the device safer for the animals, due to the exclusively low voltage provided, and easy to handle. While in our case this feature was mostly an add-on, in outdoor cages or on field research sites without direct access to power outlets, this could be

a necessary requirement. Combined with image-based animal identification[88,89], this would allow for comparative testing of captive and natural populations[90]. Finally, several structural elements of the MXBI were designed for manufacturability and commissioned to local workshops or locally 3D printed. The combination of structural and electronic hardware elements is particularly well suited, in our opinion, to replicate our device on a large scale. As a result of these built-in features, in the animal's facility of our institute, 6 devices are simultaneously active, training 12 animals in parallel over the course of several hours, and generating on average 1500 trials a day requiring only approximately 35 min of human labor.

In conclusion, all of these aspects are to be considered when establishing a successful high-throughput pipelines (across various fields of cognitive neuroscience) because together they ultimately add up to create automated high-throughput protocols for integrating advanced cognitive and behavioral assessments with physiological data recordings[38].

**Autonomous devices as cognitive enrichment**. Throughout our experiments we found that animals consistently interacted with the device regardless of their performance. In certain occasions animals performed thousands of trials at chance level, across several weeks, despite no social or fluid restriction were applied. While this might seem counterintuitive, we argue that from the animals' perspective our approach, coupled with the appeal of the liquid arabic gum that the device delivered, represents a form of enrichment[68–70]. From a psychological standpoint, cognitive enrichment strategies exercise what is known as competence, namely the range of species-specific skills animals employ when faced with various challenges. This, in turn, promotes the sense of agency, described as the capacity of an individual to autonomously and freely act in its environment[91]. Promoting both competence and agency has been proposed to be crucial for the psychological wellbeing of captive animals because: (1) animals can better cope and thus better tolerate captivity; and (2) animals can exercise species-specific cognitive abilities that have little opportunity to be expressed in captivity[68,92].

**Study limitations and caveats**. Several animals in the audio-visual association tasks performed at chance level for several thousands of trials. Receiving a reward in half of the trials might be a successful strategy for animals that are not constrained, isolated, or fluid/food restricted. Under these conditions it is unclear whether animals will attempt to maximize their reward—as has been reported in studies where food or fluid regimes are manipulated[93,94] but see[95]—or are satisfied with chance performance. An animal that is satisfied performing at chance for a certain task will naturally not 'learn' even though it might cognitively be able to. In line with this interpretation, animals that performed at chance level in a 2AC version of an auditory discrimination task, successfully performed the auditory discrimination when the overall chance level was reduced from 50 to 33% by employing a 3AC version.

Our data demonstrate flexibility of auditory training using natural stimuli and lay the groundwork for further investigations e.g. testing categorical perception of vocalizations by modulating the spectral content of the stimuli used. However, a caveat of our work is that our approaches were not successful in training marmosets on discriminating artificial sounds consistently (see supplementary materials). Among other potential explanations, we attribute this difficulty due to the introduction of auditory cues relatively late in training. This might have biased animals to focus on the visual domain—which is considered the dominant sense in primates[96,97]—while ignoring other cues. Future studies

should therefore explore alternative approaches to train arbitrary acoustic discriminations potentially by introducing reliable auditory cues very early in training.

## Methods

All animal procedures of this study were approved by the responsible regional government office [Niedersächsisches Landesamt für Verbraucherschutz und Lebensmittelsicherheit (LAVES), Permit No. 18/2976], as well as an ethics committee of the German Primate Center (Permit No. E1-20_4_18) and were in accordance with all applicable German and European regulations on husbandry procedures and conditions. It has to be noted, however, that—according to European regulations and implemented in German animal protection law — the procedures described in this study can be considered to be environmental enrichment.

**Animals**. A total of 14 adult common marmosets (*Callithrix jacchus*) of either sex (see Table 1) were involved in the experiments carried out in the animal facility of the German Primate Center in Göttingen, Germany. Some of the animals were prepared for neurophysiological and cochlear implant experiments. Animals were pair housed in wire mesh cages of sizes 160 cm (H) × 65 cm (W) × 80 cm (D) under a light-dark cycle of 12 h (06:00 to 18:00). Neighboring pairs were visually separated by opaque plastic dividers while cloths hung from the ceiling prevented visual contact across the room. Experimental sessions occurred mostly in the afternoon and without controlled food/fluid regimes or social separation from the assigned partner. Liquid arabic gum (Gummi Arabic Powder E414,1:5 dissolved in water; Willy Becker GmbH) or dissolved marshmallows (marshmallow juice, 1:4 water dilution) was provided as a reward by the touchscreen device for every correct response in the various experiments. Marshmallow or arabic gum pieces, stuck to the touchscreen, were used during the *initial training* phase.

**Apparatus**. The marmoset experimental behavioral instrument (MXBI) is directly attached onto the animals' cage and measures 44 cm (H) × 26 cm (W) × 28 cm (D). The device is internally divided into three sections (Fig. S4A). The electronics compartment on top contains: a Raspberry Pi 3B + (raspberrypi.org); a RFID module with a serial interface (Euro I.D. LID 665 Board); two peristaltic pumps (Verderflex M025 OEM Pump), one on each side; a camera module (Raspberry Pi wide-angle camera module RB-Camera-WW Joy-IT); and a power bank (Powerbank XT-20000QC3) through which 5 and 12 V (max 2.1 A) was provided to the whole system. In our setup and with our tests, the power banks last up to 8 h before the battery is depleted allowing for continuous training or testing during most of the waking hours of the colony. We chose the Raspberry Pi single board computer instead of more commonly used tablet PCs[88,98] for ease of interfacing various external devices. Towards this requirement the Raspberry Pi has various general-purpose input output capabilities allowing to integrate a wide variety of external hardware components such as microcontrollers, touchscreens, etc. with standard communication interfaces (SPI, I2C, I2S). Additionally, new MXBIs can simply be set up by copying the content of the SD card of an existing device into the SD card of the new device. The behavioral chamber in the middle (internal dimensions: 30 cm (H) × 22 cm (W) × 24 cm (D)) hosts: a 10 inch touchscreen (Waveshare 10.1"HDMI LCD [H], later sessions contained a 10" infrared touchscreen attached to the LCD screen, ObeyTec); a set of two speakers (Visaton FR58, 8 Ω, 120–20,000 Hz) for binaural acoustic stimulation; a horizontal reward tube with custom-made mouthpiece (placed at 3 cm from the screen but variable between 2 cm and 5 cm); the coil (or antenna) of the RFID and a cylindrical mesh to prevent more than one animal to be inside the device at the same time (Fig. 1a). Finally, at the bottom of the device, space is left to accommodate a removable tray to collect and clean waste. Hinges on one side allow the device to be opened from the back if cleaning or troubleshooting is needed (Fig. 1a Left). The MXBI can be anchored to the front panel of the animal's cage via custom designed rails welded to the cage. A removable sliding door at the front panel allows animals to access the MXBI when attached. A Python3 based software (Python 3.5.3 with the following modules: tkinter 8.6, numpy 1.12.1, RPi.GPIO 0.6.5, pyaudio 0.2.11) running on the Raspberry Pi records all interaction events (screen touches, RFID tag readings and video recording), manages stimulus presentation (acoustic and visual), controls the reward system and finally backs up the data automatically to a server via wireless local network connection (Fig. S4B).

**Procedure**. Behavioral training and testing sessions were started by connecting the Raspberry Pi and LCD display to power which initiates booting. After booting, a custom script with a series of preconfigured commands was automatically initiated to: (1) connect the device to a central server for automatic, recursive, data logging, as well as main database access; (2) start the local camera server for remote monitoring and video recordings (Fig. 2b); (3) automatically launch the experimental task when needed. The fluid reward was manually loaded in each device and the pump was primed. The device was then attached to the cage and the sliding door in the front panel removed for the duration of the session. At the end of the session, the sliding door was placed back between the device and the cage so that the device could be detached, cleaned, and stored. The touchscreen surface and the behavioral compartment were thoroughly cleaned to remove odors and other traces. Hot water was used daily to clean the reward system to prevent dried reward from clogging the

silicon tubes and mouthpiece. The entire process requires a single person around 35 min (15 for setting up and 20 for taking down) with six devices.

**Sessions**. In order to operate the touchscreen at the opposite end from the MXBI's entrance, the animals are required to go through the opening on the front panel and the mesh cylinder (Fig. 1a). Crossing the antenna inside the mesh cylinder identifies animals via their RFID transponder (Trovan ID-100A) implanted between the animal's shoulders for husbandry and identification reasons. Standing up inside the mesh places the animals' head 3 cm above the mouthpiece and 4–5 cm away from the screen, directly in front of a cut out in the mesh of 3.5 × 8.5 cm (HxW) through which the touchscreen can be operated (Fig. 1a). Throughout each session, animals were regularly monitored by the experimenter from a remote location (approximately every 15 min). Additionally, videos from most sessions were recorded and stored. Fluid (either water or tea) is available *ad libitum* to the animals within their home cage but outside the MXBI. Solid food was provided to the majority of the animals before, after, and during the session, depending on husbandry and/or veterinary requirements.

**Experimental paradigm**. Throughout the experiments, animals never left their home cage. With the exception of animals a and b, that where pilot subjects and underwent a different initial procedure, all animals were first trained manually to operate the device at a basic level by means of positive reinforcement training and shaping techniques (see methods section: *initial training*). Afterwards, all animals where guided by an unsupervised algorithm through a series of preconfigured training steps (see section Automated unsupervised training (AUT)) to acquire basic proficiency in a standard 2AC discrimination task. The animals' discrimination proficiency was then tested and refined in a next experiment in an acoustically guided discrimination task (see section Audio-visual association). In a third experiment, the acoustic stimuli were replaced with novel stimuli and the animal's ability to generalize was assessed (see section Generalization to novel stimuli). Last, we developed a psychoacoustic detection task to quantify the animal's hearing thresholds (see section Psychoacoustic assessment). It is important to note that not all animals took part in all experiments either because some animals were assigned to different projects or were not always available due to the requirements of different experiments.

**Initial training**. The goal of the *initial training* procedure was to instruct naïve animals to use the touchscreen. To this end, this training was divided into three sequential steps: first, habituation to the device; second, forming a mouthpiece-reward association and finally, a touch-to-drink phase. During the first two steps no wire mesh cylinder was placed inside the MXBI. Unlike the remainder of the training, all *initial training* required the constant surveillance of the experimenter, to remotely access and control the screen of the device from another computer to shape the animal's behavior while monitoring the video feed. The measured round-trip delay between observing the behavior and effectively delivering the reward was approximately 400 ms plus an additional response latency of the observer. Together, we believe that this delay should be sufficient for stimulus-response integration and association[99]. The *initial training* lasted on average 6 (±2) sessions and was routinely completed within 2 weeks. With the exception of animals a and b, all animals underwent the *initial training*.

*Device Habituation*. During this first step the device was attached to the cage without the mesh cylinder, to allow the animals to freely explore the behavioral chamber (see supplementary video 1) in sessions lasting on average 40(±20) minutes. Before switching to the next step, the experimenter ensured that both animals would show interest and no aversion towards the device (e.g. walking towards and not away from the device). The number of sessions needed to observe this behavior varied between 1 and 2.

*Mouthpiece-reward association*. Following the habituation, drops of reward of variable magnitude (between 0.3 and 0.5 ml) were remotely triggered by an experimenter in order to direct the interest of the animals towards the mouthpiece (see supplementary video 2). Presumably due to the sudden occurrence of the pump sound while rewarding, the interest towards the MXBI for some animals slightly decreased. To overcome this issue and to increase the likelihood of animals interacting with the device a number of small marshmallow pieces were placed randomly over the mouthpiece. After all pieces were consumed and the animals left the MXBI the experimenter closed the sliding door to place new pieces. Once the animals showed interest in the mouthpiece in the absence of the reward, the association was considered established and the next phase started. This step required between 1 and 5 sessions, with each session lasting 30–60 min.

*Touch-to-drink phase*. The aim of this step was to teach the animals to actively seek the reward, by triggering the touchscreen. In order to achieve such behavior efficiently and to make sure the animal used the hand and not e.g. their mouth (which was observed in pilot experiments) to touch the screen, a mesh cylinder was placed inside the device. In turn, this restricted access to one animal at a time, and improved the efficiency of the RFID identification. Additionally, small pieces of marshmallows were placed on the screen within the triggering area, to encourage

the animals to retrieve the marshmallow pieces and thereby touch the screen. When all pieces were consumed and the animal had left the MXBI the experimenter closed the sliding door to place new pieces on the screen and resumed the session. While the marshmallow pieces where collected, fluid reward was provided, triggered either remotely by the experimenter or by the animals themselves touching the stimulus on the screen. This procedure successfully allowed all animals to switch from reaching to retrieve the marshmallows to simply touching the screen to trigger fluid reward (see supplementary video 3.1 and 3.2). After 5–10 consecutive reaching movements towards the screen in the absence of marshmallows, the behavior was considered acquired and the *initial training* concluded. Between 1 and 4 sessions (each lasting 60 min on average ± 10 min) were necessary to finish the touch-to-drink phase.

**Automated unsupervised training (AUT)**. Upon completion of the *initial training* phase, all animals underwent an automated stepwise protocol designed to gradually bring the animals from a *quasi*-naïve state to proficiency in a 2 alternative-choice (2AC) audio-visual association task. Throughout the protocol the performance of the animal was constantly monitored by an algorithm to adjust the task difficulty, by changing parameters as well as introducing or removing elements in the training (Fig. S4C). Animals ascended in steps by performing at least 8 trials out of 10 correctly and descended when less than 3 trials out of 10 were correct (Fig. S4D). Finally, the progress of each animal was automatically stored and retrieved on each trial, so that the animals could navigate the series of steps and resume from the last step they were in when they left, across breaks and sessions. The automated training protocol (AUT[48]) was comprised of 48 steps, grouped into four milestones: decrease of the size of a visual stimulus (trigger) to be touched for reward; change of position of said stimulus; introduction of sound and delayed presentation of a visual target; introduction of a second visual target as a distractor. Through these steps and milestones, the animals were trained on the basics of how to operate a touchscreen within the context of a standard 2AC visually guided task. The aim of the AUT was to prepare the animals for an audio-visual association experiment, in which they were required to distinguish between different sounds by selecting a corresponding visual stimulus. During the first 15 steps (size milestone) a white circle embedded in a blue rectangle (trigger) placed on the vertical meridian had to be touched to obtain a reward (0.1–0.2 ml). From step 2 to 15 the trigger gradually shrunk in size from 6 × 6 cm to the final size of 3 × 3 cm. Touching the screen outside the trigger resulted in a 2.5 s (earlier sessions) to 5 s (later sessions) timeout indicated by a gray screen during which no new trial could be initiated and touches were ignored. A touch within the boundaries of the trigger resulted in reward administration (as above), followed by a new trial which could be started after 0.8–2.5 s. In steps 16–30 (position milestone) the trigger's position gradually changed by 5 mm at each step, either to the left or to the right of the original central position, until the edge of the screen was reached. From step 31 onwards (delay milestone) the trigger first appeared at the center and upon touch reappeared at the left or the right edge of the screen and had to be touched again. The reward was delivered if both touches were executed correctly. Only touching outside of the second trigger resulted in a timeout. This was done to ease the transition from one stimulus to two different stimuli presented, which was occurring on all steps starting with step 36. Throughout these steps the second trigger was replaced randomly with one of two visual stimuli (targets): either the picture of an infant marmoset face (3 × 3 cm), or an abstract geometric pattern (3 × 3 cm) (Fig. 3a). Starting from step 36 an acoustic stimulus (either a repeated infant marmoset vocalization[100]; or a train of pure tones—sTr—chosen for individual animals from a range between 1.5 and 3.5 kHz) was presented 1–1.5 s before the visual target, with a gradually increasing sound intensity (in steps of 10 dB) from 32 ± 2 dB SPL on step 36 to a final loudness of 72 ± 2 dB SPL on step 40. The vocalization was followed by the marmoset face (for 5 s) while the sTr was paired with the geometric pattern (Fig. 3a). From step 41 to step 45 the parameters were kept the same as step 40, to provide prolonged and unchanged exposure to the visuo-acoustic stimulus. At step 46 (visual 2AC milestone) a visual distractor was displayed together with the target but on the opposite edge of the screen. In the case of a 'vocalization' trial the visual distractor was the geometric pattern and vice versa. The distractor was gradually increased in size from 0.3 × 0.3 cm on step 46 to 2.8 × 2.8 cm on step 49. Thus, from step 46 to 49, animals could exploit the size difference between the visual target and distractor to respond correctly by choosing the larger visual stimulus. Throughout the protocol, if no response was observed within 7 s from stimulus presentation the trial was aborted and the trial outcome was labeled as 'ignored'. The AUT described here (version #10) is the result of several attempts that are described in the supplementary material (Table S2).

**Audio-visual association**. The audio-visual association experiment starts when an animal reaches step 50 in the Autonomous Training (Fig. 3). Contrary to the AUT, no visual cue could be used to correctly identify the target of a given trial. Here animals had to solely rely on auditory cues to obtain a reward above chance level. In this experiment, no AUT algorithm was employed and therefore the trial structure and sequence remained unchanged throughout. This experiment consisted of a two-alternative choice task (2AC), where only one of the two available options was the correct one and the animal's ability to distinguish the options was assessed from the animal's relative frequency of choice. We implemented two variants of this task, a 2AC and a 3AC, plus a control condition (see supplementary material). Both variants employed the same stimuli of the Autonomous Training with added visual

distractors in the 3AC variant which had no sound associated and were not presented as target, but always as a distractor. While touching the target of a given trial was rewarded, touching a distractor resulted in 5 s (later sessions) timeout indicated by a gray screen during which no new trial could be initiated and further touches were ignored. On the contrary, after a correct response, a new trial could be started 0.8–2.5 s after reward delivery. A detailed timeline of an example trial from this task is shown in Fig. S4E and a video of an animal performing few trials in the 2AC variant is available in the supplementary materials (Video 4). Animals who did not perform above chance on the 2AC variant were assigned to the 3AC variant. The 3AC variant was used to lower the chance of obtaining a reward randomly at any given trial from 50% to 33%. Two animals who performed at chance level in the 2AC were assigned to a different experiment and could not be tested on the 3AC.

**Generalization to novel stimuli**. To evaluate the flexibility of our protocols and determine whether the animals could generalize the already learned 2AC task using different sounds, we performed 4 different variations of the already described 2AC task. Here we tested a twitter vs a pure tone, a phee vs a pure tone, a twitter vs white noise and an infant vocalization vs a twitter. To avoid a high number of changes within every task switch, once the animals learned the first task variation (twitter vs pure tone) they were always brought back to this task to stabilize their performance before moving to the next task variation. Vocalizations were recorded from a different colony[101]. Representative visual indicators that matched every single acoustic stimulus are shown in Fig. 4.

**Psychoacoustic assessment**. In order to assess the animals' hearing thresholds, we devised a simple detection task based on the discrimination task used before. In this task animals were trained to choose the gray triangles (previous visual stimulus of sTr) to report the absence of the vocalization (i.e. silence). Once the behavior was stable (after two sessions) and based on the measured background noise of the facility (60 ± 5 dB SPL, see below and Fig. 5c) we set the sound intensities to 0, 15, 30, 45, 60, 70, 80db SPL for the vocalization. Given that some of these intensities were below the background noise of the facility, all trials with intensities between 15 and 45 dB SPL were rewarded regardless of the choice of the animal (Fig. 5a). Moreover, vocalization trials at 0 dB SPL were rewarded if the triangles were selected (visual stimulus for the silence). This was instrumental to first account for both type of trials (silence and vocalization) presented at 0 dB SPL, and second to effectively establish the task as a detection rather than a discrimination task. Finally, all sessions were performed in the afternoon, from 1 pm to 4.30 pm, when the colony's background noise was the lowest with feeding and personnel's activity occurring mostly in the morning.

In order to measure the background noise level of the facility inside the MXBI a microphone (Bruel And Kjaer Type 4966 1/2-inch) was placed at the marmosets' ear level and a measuring amplifier (Bruel And Kjaer Measuring Amplifier Types 2610) visualized the sound pressure level. The sound output of the two devices used to gather hearing thresholds (1 for animals a & b, 1 for animal d) were further calibrated inside an insulated sound proof chamber. An amplifier (Hifiberry amp2) coupled to the Raspberry Pi produced the audio signal, while a measuring amplifier (Bruel And Kjaer Measuring Amplifier Types 2610) and a microphone (Bruel And Kjaer Type 4966 1/2-inch) placed at the marmoset ear level pointing towards one speaker, acquired the sound output. Additionally, an oscilloscope (Rigol DS1000Z), attached to the output lines of the amplifier, measured the voltage. We were able to corroborate the step size (0.5 dB SPL) of the amplifier by sampling 5 different frequencies (0.875 kHz, 1.75 kHz, 3.5 kHz, 7 kHz, 14 kHz) at 10 different sound pressure levels (100 dB, 95 dB, 90 dB, 85 dB, 80 dB, 75.5 dB, 70 dB, 65.5 dB, 60 dB, 50 dB). We found a stable and accurate correspondence between the values provided to the amplifier, the sound pressure levels measured by the measuring amplifier, and the voltage values measured by the oscilloscope.

**Data treatment and Statistics**. Data acquisition, processing, analysis, and statistical testing were performed in Python 3.5.3 and 3.9. Statistics and significance tests for Figs. 1–4 were calculated via the packages *scipy*[102,103] and *numpy*[104], co-installed upon installation of the package seaborn. An alpha level of less than 0.05 was considered significant. Data formatting and visualization for the same figures as well as for Table 1 was achieved with the packages *pandas*[105] and *seaborn* (seaborn.pydata.org). Hit rate's significant difference from chance (Fig. 3c) was assessed with a Binomial test; while reaction time differences between the two presented auditory stimuli (Fig. 3d) were tested for significance with a Kruskal-Wallis test by ranks. Both tests were adjusted post-hoc for multiple comparisons with Bonferroni correction (corrected alpha = 0.0019, from the python module *statsmodel.stats.multitest.multipletests*). In Fig. 2d and Fig. 3a, b, the variable "percentage of trials" on the abscissa was used to achieve a shared and standardized axis on which multiple animals could be compared and visualized against each other, irrespective of the total amount of trials each individual performed. The assumption behind this choice was that learning occurs through similar mechanism across individuals, but unfolds through a different amount of trials that depend on each animal's engagement level. The resulting process of standardization attenuated the inter-individual variability between animals for parameters such as steps of the AUT (Fig. 2c) and Hit Rate (Fig. 3a, b, and 4b).

Psychometric function estimation was achieved with the python module *psignifit*[106] set to fit a cumulative normal sigmoid function, with all parameters free

and with 95% confidence intervals. The resulting function can be expressed as follows:

$$\psi(x;m,w,\lambda,\gamma) = \gamma + (1 - \lambda - \gamma)S(x;m,w) \qquad (1)$$

where $m$ represents the threshold (level at 0.5), $w$ represents the width (difference between levels 0.5 and 0.95), $\lambda$ and $\gamma$ represent the upper and lower asymptote respectively (Eq. (1) in ref. [106]).

**Reporting summary**. Further information on research design is available in the Nature Research Reporting Summary linked to this article.

## Data availability
The datasets[107] generated during and/or analyzed for the current study are available at GitHub (https://github.com/CHiP-Lab/mXBI) and Zenodo (https://doi.org/10.5281/zenodo.6139297).

## Code availability
The code[107] to recreate the data figures are available at a dedicated Github repository (https://github.com/CHiP-Lab/mXBI) and Zenodo (https://doi.org/10.5281/zenodo.6139297).

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

## Acknowledgements
The authors would like to thank Dr. Steffen Hage and Dr. Michael S. Osmanski for providing juvenile and adult call exemplars, respectively. Additionally, we would like to thank Karin Tilch and Manfred Eberle of the German Primate Center for the pictures of the device and marmosets, respectively, used in this manuscript. The team around R. Schürkötter of the workshop of the Max-Planck-Institute for Multidisciplinary Sciences manufactured the device cases. The work was funded by the European Research Council (ERC) under the European Union's Horizon 2020 research and innovation program (grant agreement No 670759– advanced grant "OptoHear" to T.M.), by the Deutsche Forschungsgemeinschaft (DFG, German Research Foundation) via the Leibniz Program (MO896/5 to T.M.), and by the Max-Planck-Society (fellowship to T.M.). In addition, work by T.M. was supported by Fondation Pour l'Audition (FPA RD-2020-10).

## Author contributions
A.C., M.J. conceived the study. A.C., J.C.M., M.J. designed and developed hardware and software. J.C.M. performed the experiments with the help of M.J. J.C.M. curated the data and A.C. analyzed the data and generated the figures. A.C., J.C.M., M.J. discussed and interpreted the data. T.M. provided funding, discussions and feedback on all aspects of the study. M.J. supervised the study. A.C., J.C.M., M.J. wrote the paper with input from T.M.

## Funding

## Competing interests
The authors declare no competing interests.
