## [Peer review file · Nature Communications]

Reviewers' comments:

Reviewer #1 (Remarks to the Author):

The study of Calapai and colleagues is part of a series of studies in which they develop and use automated cage-based training systems for nonhuman primates. Three previous reports described the technology and the training approaches pertinent to macaque monkeys, particular for testing visual perception. The present manuscript describes respective aspects pertinent to marmoset monkeys and focusing on audiological testing. For this species, such systems have already been used for testing visual perception (e. g. Spinelli et al. 2004).

The choice of this species was motivated because marmosets have recently become an important genetically tractable primate animal model for different research questions, requiring tools to effectively test large numbers of animals. Marmosets are also considered a promising model system for studies of auditory and vocal processing. In the manuscript, the authors describe a system that appears to be robust and capable of training relatively large numbers marmosets with significantly less time efforts than needed with traditional lab-based procedures. Further the new device has the advantage that it does not require fluid or food restrictions or social separation. The authors also describe shaping procedures that appear to ease auditory training. It is not clear, however, why only a fraction of the monkeys participating in this study learnt to successfully to perform the audiological tests. It is also not clear whether the automated training device allows to assess auditory cognitive abilities in this species other than those that have been tested with traditional training procedures (e. g. Osmanski et al., 2016).

Reviewer #2 (Remarks to the Author):

In this study, Calapai et al. developed a cage-based training platform for the marmoset monkey. In comparison with the similar devices developed previously, this one has two key features: 1) optimized for auditory task training and 2) using an automated, unsupervised training method. In general, I am supportive of tool development to facilitate our research efficiency and also to refine our non-human primate research in accordance with 3Rs. However, given previous publications for cage-based training systems even for the marmoset (e.g., Takemoto et al., 2011), the novelty of this research would be not strong enough to be published in Nature Communications. Also the focus of this study is relatively narrow, and so I am afraid that the broad readers may not be interested in this topic.

Major point

1. Although I appreciate the authors carefully describing the details, too many details mask major findings and what the authors wanted to emphasize. So it would be better to write Results and Discussion concisely.

2. Five of nine tested monkeys showed differences in the reaction time depending on the stimulus type (longer RTs for vocalizations). Could the authors explain the reason why?

3. How is the distribution of inter-trial intervals for successive correct trials? The authors used the timeout after error trials, but if monkeys do not continuously engage the task, the timeout would not facilitate learning.

Minor point

1. How did the author determine the length and end of each session?

2. D-prime would be more appropriate than just reporting the hit rate.

Reviewer #3 (Remarks to the Author):

This study develops a system to train marmosets for visual/auditory tasks while animals remain in home cages and shows progress of the animals' task performance. While in-cage training becomes increasingly popular, those for auditory tasks is scarce except for a few (e.g. Archakov et al., 2020 PNAS). NHP tend to have visual biases during sensory tasks, as acknowledged by authors in Discussion. The manuscript describes in detail how to advance task parameters, starting with visually-guided tasks, to get marmosets engaged in auditory tasks eventually.

While the system is portable and its actions are programmable, most of it appears to adopt basic setup for visual tasks and modify it to add auditory stimulation.

Tasks appear mostly visually-guided. Consistent success of training animals to make auditory decision was shown only for a matching a single pair of one conspecific vocal sound and one marmoset face photo.

The system described in this study will be appreciated if authors describe its more general applicability, e.g. how it can be used for auditory psychophysics.

I have several concerns.

(1) While both the hardware and software functioned reliably as they were used for multiple animals and many times, training protocols to train animals to perform auditory tasks seem still at a pilot experimental stage and not well established.

(2) The manuscript describes that sound intensity was increased with difficulty steps. I speculate that starting from soft sound is good for habituation. However, softer sound would be rather difficult to detect/discriminate than loud sound in general. So, to train animals to pay more auditory attention, it may be needed to decrease, rather than increase, the sound level with difficulty.

(3) Besides vocal sound and face photo and octave separation of tones for complex tone repetition, both visual and auditory selected stimuli seem too arbitrary. I am not sure if it was good idea to keep using unsuccessful tones trains for all animals.

Even though marmosets can discriminate the pitch of tones used in this study, the steps taken to reach the level requiring pitch discrimination do not seem to help animals to discriminate. Also, training steps instructed animals to discriminate only sizes of visual stimuli regardless of visual stimulus types that were meaningless. Additional physical features that change systematically or gradually with pitch (e.g. shape, hue) could be helpful. Similarly, for auditory stimuli, tone vs. noise, frequency modulation directions. Even though temporal patterns of vocal sound and tone trains were very different, temporal structures of visual stimuli were not. It may be helpful for animals if visual target blinked at same rate as tone train and visual distractor blinked at different rate as well. By considering those possibilities, I am not sure to conclude that maneuvers used in this study were enough to train marmosets to learn auditory tasks.

(4) Each daily session of AUT started from step 1 and ends by completing the last step as shown in Fig. 2A. Is it correct? If yes, a sentence stating that would be appreciated, because behavioral tasks for other species (i.e. macaques and humans) usually do not step up difficulties within blocks.

Were there no days that animals did not advance to the last step during a session?

Even though marmosets may need many steps to perform the goal tasks daily, I am not sure whether so many steps, particularly of changes in visual cues, are needed to learn every time. Do marmosets forget tasks every day?

(5) 14 animals in this study were pair housed. However, the numbers of sessions in Table 1 do not appear as pairs. Were there pairs in which only one of them performed the tasks?

(6) Lines 682-685 "Here, animals were observed to weakly reach, ..." Here describes failure of sensor detection lead animals to touch other objects as an alternative strategy of marmosets. If a sensor to

detect animal's action is not reliable, is it better to provide visual feedback for correct touches detected by touch pad sensor (e.g. flash in the surrounding of visual target) in addition to a reward?

(7) Lines 330-331 "along the horizontal upper midline of the screen" What is meant here is not clear.

(8) Line 428 typo "animals animals"

Reviewer #1 (Remarks to the Author):

The study of Calapai and colleagues is part of a series of studies in which they develop and use automated cage-based training systems for nonhuman primates. Three previous reports described the technology and the training approaches pertinent to macaque monkeys, particular for testing visual perception. The present manuscript describes respective aspects pertinent to marmoset monkeys and focusing on audiological testing. For this species, such systems have already been used for testing visual perception (e. g. Spinelli et al. 2004).

While some studies have already used cage based devices for studying visual perception and cognition based on the visual domain, we argue that the major advance of our system and training is the focus on the auditory domain. This advance is particularly relevant as monkeys have been notoriously difficult to train in the auditory domain, and generally display a bias towards vision. E.g. it has been shown that baboons can easily learn to locate food items based on visual but not auditory cues (Schmitt & Fischer, 2009). Among other results this failure at such a seemingly simple auditory task has led to the suggestion that inferential reasoning is modality specific. Maybe as a result, one observes that studies on testing auditory tasks in NHPs have remained quite crude in comparison to those in the visual domain and almost exclusively employ Yes/No or Go/NoGo type tasks. Yes/No and Go/NoGo type tasks are prone to changing response bias as they are very likely to occur especially during learning and training. These caveats can be circumvented by task designs that balance these effects across stimuli. This was our original driving force to develop a two alternative-choice design (2AC) akin to classical two alternative-forced choice designs. We consider this one of the major advances provided by our manuscript.

The choice of this species was motivated because marmosets have recently become an important genetically tractable primate animal model for different research questions, requiring tools to effectively test large numbers of animals. Marmosets are also considered a promising model system for studies of auditory and vocal processing. In the manuscript, the authors describe a system that appears to be robust and capable of training relatively large numbers marmosets with significantly less time efforts than needed with traditional lab-based procedures. Further the new device has the advantage that it does not require fluid or food restrictions or social separation. The authors also describe shaping procedures that appear to ease auditory training. It is not clear, however, why only a fraction of the monkeys participating in this study learnt to successfully to perform the audiological tests.

This is a finding from our work that we also pondered about substantially. While it remains speculation, it is our belief that all tested animals are cognitively capable of performing the task. We then attribute the failure to perform successfully to multiple factors. In traditional psychophysical experiments with monkeys the motivation is manipulated by food or fluid regimes. This is to ensure that animals will participate in the experiments and attempt to optimize their performance. In our experiments, motivation is not manipulated externally and therefore needs to be internal. In this case, it is unclear whether animals will attempt to maximize their reward or are satisfied with chance performance. An animal that is satisfied performing at chance for a certain task will naturally not 'learn' even though it might cognitively be able to. We suggest that this is indeed the case for animals that did not learn, i.e. all animals are cognitively capable of discriminating sounds by picking a certain visual response indicator. An additional experiment which we performed in response to the reviewer's comment addressed this question. We reasoned that a reduction of chance level might lead to increased pressure to follow the task rules and increase the number of animals successfully performing. Therefore, we devised a 3 alternative-choice (3AC) version of the

task. Here, 4 out of 4 previously unsuccessfully performing animals learned the task (animals k and j – who had failed the natural discrimination task; and animals g and h who had failed the artificial discrimination task). Together, now 9 out of 11 animals learned the task. The remaining two animals that did not learn the 2AC discrimination were assigned to a different project and could not be tested on the 3AC version.

It is also not clear whether the automated training device allows to assess auditory cognitive abilities in this species other than those that have been tested with traditional training procedures (e. g. Osmanski et al., 2016).

In the revised and extended manuscript, we added two additional experiments that further enhance the contribution of our approach. First, in order to test for flexibility of training and learning as well as to lay the groundwork for assessing e.g. categorical perception of vocalization types, we trained new pairs of acoustic stimuli with our self-paced, home-cage based device (termed generalization experiment in the revised MS). Second, we demonstrate psychometric evaluation of marmosets hearing threshold of complex stimuli (termed hearing threshold experiment in the revised MS) in a more naturalistic, i.e. vocal, environment in comparison to the typical chair-based psychophysics e.g. in Osmanski et al. 2016.

As pointed out throughout the revised “Discussion” section, besides the ethical implications of conducting psychoacoustics experiments without social nor dietary restrictions; these two aspects show: 1) a high degree of cognitive flexibility of marmoset monkeys in the hearing domain that was not reported before; and 2) suggest that psychoacoustic assessment in a more naturalistic environment can reveal cognitive hearing abilities of the animals in complex listening situations.

Reviewer #2 (Remarks to the Author):

In this study, Calapai et al. developed a cage-based training platform for the marmoset monkey. In comparison with the similar devices developed previously, this one has two key features: 1) optimized for auditory task training and 2) using an automated, unsupervised training method. In general, I am supportive of tool development to facilitate our research efficiency and also to refine our non-human primate research in accordance with 3Rs. However, given previous publications for cage-based training systems even for the marmoset (e.g., Takemoto et al., 2011), the novelty of this research would be not strong enough to be published in Nature Communications. Also the focus of this study is relatively narrow, and so I am afraid that the broad readers may not be interested in this topic.

We would like to thank the reviewer for the constructive and critical assessment of our work. We have addressed these two very important issues in the Result and Discussion sections of the revised manuscript where we present and elaborate on additional experiments and data that broaden the scope of the study, at least in our view.

While some studies have already used cage based devices for studying visual perception and cognition based on the visual domain, we argue that the major advance of our system and training is the focus on the auditory domain. This advance is particularly relevant as monkeys have been notoriously difficult to train in the auditory domain, and generally display a bias towards vision. E.g. it has been shown that baboons can easily learn to locate food items based on visual but not auditory cues (Schmitt & Fischer, 2009). Among other results this failure at such a seemingly simple auditory task has led to the suggestion that inferential reasoning is modality specific. Maybe as a result, one observes that studies on testing auditory

tasks in NHPs have remained quite crude in comparison to those in the visual domain and almost exclusively employ Yes/No or Go/NoGo type tasks. Yes/No and Go/NoGo type tasks are prone to changing response bias as they are very likely to occur especially during learning and training. These caveats can be circumvented by task designs that balance these effects across stimuli. This was our original driving force to develop a two alternative-choice design akin to classical two alternative-forced choice designs. We consider this one of the major advances provided by our manuscript.

To expand the focus of our work we now describe two new types of experiments: one to demonstrate that our device would allow for auditory psychophysics (termed hearing threshold experiment in the revised MS) and second to test for flexibility of training by testing various combinations of both, different natural stimuli and arbitrary stimuli (termed generalization experiment in the revised MS). As a straightforward and hitherto not described psychophysical assessment, we chose to study thresholds for detection of a spectro-temporally complex sound – in this case a juvenile vocalization. All 3 animals tested reliably engaged in the detection task and allowed us to gather detailed psychometric functions with thresholds of 37, 39 and 47 dB SPL. This further demonstrates that our device allows to study processing of sounds by common marmosets under naturalistic, potentially distracting conditions. Additionally, we could show that marmosets can flexibly discriminate natural from artificial stimuli with a hit rate of higher than 75 %: i.e. regardless of the vocalization chosen as the natural stimulus they can discriminate from a tone or noise. The data therefore demonstrate flexibility of auditory training and lay the groundwork for further investigations. Future work could e.g. test categorical perception of vocalizations by modulating the spectral content of the stimuli used. Together, these data open up a whole new avenue to study auditory cognition in primates. We believe that our work is therefore of interest to a broad audience such as targeted by Nature Communications.

Major point

1. Although I appreciate the authors carefully describing the details, too many details mask major findings and what the authors wanted to emphasize. So it would be better to write Results and Discussion concisely.

We have revised both the structure and the content of the manuscript to address this very important concern. The manuscript now features a streamlined description of the developed methodology, a concise description of the results; as well as a restructured discussion section in which the major findings are discussed first and put in a more general context at the end. Together, we are confident that the revised manuscript is more accessible to the general reader.

2. Five of nine tested monkeys showed differences in the reaction time depending on the stimulus type (longer RTs for vocalizations). Could the authors explain the reason why?

While we find this point particularly interesting and worth speculating, we currently have no clear explanation for the differences in reaction time. We assessed reaction times to evaluate whether or not the animals processed the two stimuli in a different way. This would provide corroborating evidence towards the acquisition of the acoustic discrimination. One possibility could be a difference in sound level between the pure tones vs. vocalizations. We matched the peak sound levels as explained in the methods but since the vocalizations contain amplitude modulations (i.e. increase in sound level to a maximum roughly in the middle) it is conceivable that animals needed more time to assess the identity of the vocalizations. Another possibility could be the informational content of the vocalization. For the original design of our task we qualitatively assessed the acoustic structure within the colony. In this colony no

breeding pairs and therefore no baby or juvenile animals were housed. To achieve maximal separability of the presented sound and the acoustic environment, we therefore chose a vocalization type that does not occur in the colony itself. This might have led to our animals listening longer to the vocalization.

3. How is the distribution of inter-trial intervals for successive correct trials? The authors used the timeout after error trials, but if monkeys do not continuously engage the task, the timeout would not facilitate learning.

For the initial submission we did not conduct a detailed analysis of inter-trial intervals as our observations from the video stream suggested that animals quickly learn that during the timeout no new trials can be performed. Usually, animals react to a timeout in one of two ways: 1st animals either crouch down in the mesh tube to leave the device or 2nd animals turn away from the touch screen and look outside the device through slots in the side panel (used to adjust the feeding tube to screen distance). In contrast, after correct trials animals remain facing the touch screen and usually start another trial. We have now performed a more systematic analysis of inter-trial intervals that we added to the supplementary materials – Figure S3 and Table S2. The analysis revealed shorter average ITIs after correct vs. wrong trials suggesting an effect of timeouts on animal behaviour.

Minor point

1. How did the author determine the length and end of each session?

As all experiments were performed within an experimental colony with various experimental projects and husbandry as well as veterinary requirements, we adopted our experiments around these requirements first. Initially, all sessions were started in the afternoons (around 03:00 PM), during which the facility was least used by animal care takers and other experiments. In later sessions, we also started experimental sessions during the care takers break which mostly commenced around 10:30 AM. All sessions were manually ended close to the facility “lights off” time (05:00 PM). The sessions which were used to gather hearing thresholds were conducted from 2:30PM to 5:00PM, the time period were the facility was least used by others.

2. D-prime would be more appropriate than just reporting the hit rate.

We would like to thank the reviewer for suggesting the use of D-prime rather than just reporting hit rates. Initially, we did not calculate D-primes as in our task design the hit rate can be calculated as $HR = 1 - FA$: the animals have to always choose between 2 visual response indicators. While ignored trials (which were really rare, usually much less than 1 % of trials) were not included in the analysis, a bias to choose e.g. preferentially one side of the touchscreen or one visual response indicator would lead to evenly distributed mistakes. This was one of the reasons that motivated us to choose the task design in the first place. Taking this into account the D-prime can be calculated as $1/\sqrt{2} * (Z(HR) - Z(FA))$. We have now also provided D-prime values in the figures.

Reviewer #3 (Remarks to the Author):

This study develops a system to train marmosets for visual/auditory tasks while animals remain in home cages and shows progress of the animals' task performance. While in-cage training becomes increasingly popular, those for auditory tasks is scarce except for a few (e.g. Archakov et al., 2020 PNAS). NHP tend to have visual biases during sensory tasks, as acknowledged by authors in Discussion. The manuscript describes in detail how to advance task parameters, starting with visually-guided tasks, to get marmosets engaged in auditory tasks eventually.

While the system is portable and its actions are programmable, most of it appears to adopt basic setup for visual tasks and modify it to add auditory stimulation.

Tasks appear mostly visually-guided. Consistent success of training animals to make auditory decision was shown only for a matching a single pair of one conspecific vocal sound and one marmoset face photo.

The system described in this study will be appreciated if authors describe its more general applicability, e.g. how it can be used for auditory psychophysics.

We have addressed this important concern in the revised version of the manuscript in depth. To further enhance the contribution of our work we now describe two new types of experiments: one to demonstrate that our device allows for auditory psychophysics (termed hearing threshold experiment in the revised MS) and a second one to test for flexibility of training by testing various combinations of both, different natural stimuli and arbitrary stimuli (termed generalization experiment in the revised MS). As a straightforward and hitherto not described psychophysical assessment, we chose to study thresholds for detection of a spectrotemporally complex sound – in this case a juvenile vocalization. All 3 animals tested reliably engaged in the detection task and allowed us to gather detailed psychometric functions with thresholds of 37, 39 and 47 dB SPL. This further demonstrates that our device allows to study processing of sounds by common marmosets under naturalistic, potentially distracting conditions. Additionally, we could show that marmosets can flexibly discriminate natural from artificial stimuli with a hit rate of higher than 75 %: i.e. regardless of the vocalization chosen as the natural stimulus they can discriminate from a tone or noise. The data therefore demonstrate flexibility of auditory training and lay the groundwork for further investigations. Future work could e.g. test categorical perception of vocalizations by modulating the spectral content of the stimuli used. Together, these data open up a whole new avenue to study auditory cognition in primates. Therefore, we consider our study to be of interest to a broad audience.

I have several concerns.

(1) While both the hardware and software functioned reliably as they were used for multiple animals and many times, training protocols to train animals to perform auditory tasks seem still at a pilot experimental stage and not well established.

We agree with the reviewer that the presented multitude of failed attempts and success at training of a fraction of tested animals can be taken to consider this a preliminary/ 'pilot' stage. However, we argue that the work actually goes well beyond this stage. Our attempts which are based on a large breadth of training approaches that were successful in rodents and ferrets (see also our response to point 3) and included hundreds of trials with numbers of animals which are higher than most primate studies suggest that with our failed attempts we did present negative results. It was our intention to also detail these approaches to aid work by

others which came at the cost of clarity of our main points. We have now fundamentally rewritten the manuscript to focus on our main points.

Additionally, we present further experiments increasing the success rate of training, reporting on animals' discrimination flexibility and adding substantial data regarding the animals' acoustic thresholds. It should be noted, however, that in our experiments the animal's motivation to engage is not manipulated externally (via fluid/food control or social separation) and therefore needs to be internal. In this case, it is unclear whether animals will attempt to maximize their reward - as is reported in studies where food or fluid regimes are manipulated - or are satisfied with chance performance, which is possible with 2AFC. An animal that is satisfied performing at chance for a certain task will naturally not 'learn' even though it might cognitively be able to. Therefore, we caution that it might not be expected that all animals can be 'trained' with a 2AC protocol. In fact, in the newly performed 3AC experiments we demonstrate more robust learning: 4 of 4 animals that did not learn the task with 2AC before, were successfully trained with the 3AC task.

(2) The manuscript describes that sound intensity was increased with difficulty steps. I speculate that starting from soft sound is good for habituation. However, softer sound would be rather difficult to detect/discriminate than loud sound in general. So, to train animals to pay more auditory attention, it may be needed to decrease, rather than increase, the sound level with difficulty.

This point is well taken. In the initial phases of the project we focussed on finely introducing the sound to avoid startling the naïve animals. This was introduced – in early automated, unsupervised training versions – as we have observed animals to leave the device as soon as the sounds were played the first times. While this strategy might have worked to avoid this response, we agree with the reviewer that at the same time it might have impaired the auditory association.

(3) Besides vocal sound and face photo and octave separation of tones for complex tone repetition, both visual and auditory selected stimuli seem too arbitrary. I am not sure if it was good idea to keep using unsuccessful tones trains for all animals. Even though marmosets can discriminate the pitch of tones used in this study, the steps taken to reach the level requiring pitch discrimination do not seem to help animals to discriminate. Also, training steps instructed animals to discriminate only sizes of visual stimuli regardless of visual stimulus types that were meaningless.

We are grateful for these suggestions. We agree that from the very beginning our automated unsupervised training (AUT) was strongly biased towards the visual domain and the new AUT procedures that we plan on exploring will be more balanced. We agree and are confident that such balance, or even a slight bias towards acoustic cues will push the animals to rely more on the auditory information.

Additional physical features that change systematically or gradually with pitch (e.g. shape, hue) could be helpful. Similarly, for auditory stimuli, tone vs. noise, frequency modulation directions. Even though temporal patterns of vocal sound and tone trains were very different, temporal structures of visual stimuli were not. It may be helpful for animals if visual target blinked at same rate as tone train and visual distractor blinked at different rate as well. By considering those possibilities, I am not sure to conclude that maneuvers used in this study were enough to train marmosets to learn auditory tasks.

One of the findings from our work that we consider especially noteworthy is how difficult it indeed was to have marmosets associating stimuli across the auditory and visual modality. For our first AUT versions which employed only artificial sounds we considered successful concepts from training of rodents and ferrets. The guiding strategy was always that additional properties of the stimuli should match across modalities to support crossmodal association. E.g. we presented auditory and visual stimuli together with the reward, or timeout screen, in a temporally overlapping fashion which leads to strong associations of stimulus components in rodents. Also, we presented the sound from the speaker on which side the correct visual response indicator was located. This e.g. has been shown to be an extremely strong cue for ferrets to guide choice towards the respective sound direction. In stark contrast, none of these approaches were successful in marmosets. The reviewer's suggestion of using a common temporal structure between visual and acoustic stimuli is among the potential further possibilities to match stimuli along more stimulus dimensions which we discussed during designing further AUTs. Work by the Groh lab (Mohl et al. 2020, <https://doi.org/10.1152/jn.00046.2020>) demonstrated recently that creating a common audiovisual object is linked to spatial separation of visual and auditory stimuli and fails with separations of larger than 12 degrees. As in our device the spatial separation is much larger, we did not attempt to test whether rhythmical cues could aid training and overcome the spatial separation concern. Rather, we resorted to the naturalistic approach that is described in our manuscript. While we do not think that this is the only approach possible, we believe that it is – to the best of our knowledge – the first: 1) to train marmoset monkeys on auditory tasks without requiring fluid/food restriction nor social isolation; 2) implementing auditory driven 2 alternative choice tasks; 3) that allows for flexible discrimination of various auditory stimuli, and 4) that allows to conduct home-cage based auditory psychophysics. Nonetheless, it is an interesting question in cognitive hearing whether primates can transfer knowledge from the auditory to the visual modality based on temporal patterns as has been presented in preliminary fashion for Mongolian gerbils (Orabona et al. 2009, <https://ieeexplore.ieee.org/document/5175515>). Future experiments could employ our device as well as the presented approach to study these very processes.

(4) Each daily session of AUT started from step 1 and ends by completing the last step as shown in Fig. 2A. Is it correct? If yes, a sentence stating that would be appreciated, because behavioral tasks for other species (i.e. macaques and humans) usually do not step up difficulties within blocks.

Were there no days that animals did not advance to the last step during a session?

Even though marmosets may need many steps to perform the goal tasks daily, I am not sure whether so many steps, particularly of changes in visual cues, are needed to learn every time. Do marmosets forget tasks every day?

We would like to thank the reviewer for pointing out that our description can lead to misunderstandings. In each daily session the AUT did not restart from step 1. Rather, the last step that an animal was in was loaded upon entering the device each time an animal accessed the device. This was independent of whether the entrance happened on the same session or in a next session on a separate day. This allowed animals to go through the AUT slowly and at their own pace independent of the number of trials an animal performs each session.

We have now rewritten our description (line 823 ff) that the AUT did not start every day from step 1, but instead the animal's progress was saved and resumed from session to session:

“Throughout the protocol the performance of the animal was constantly monitored by an algorithm to adjust the task difficulty, by changing parameters as well as introducing or removing elements to/from the training (Figure 6A). Animals ascended in steps by

performing at least 8 trials out of 10 correct and descended in steps when less than 3 trials out of 10 were correct (Figure 6B). Finally, the progress of each animal was automatically stored and retrieved on each trial, so that the animals could navigate the series of steps and resume from the last step they were in when they left, across breaks and sessions.”

(5) 14 animals in this study were pair housed. However, the numbers of sessions in Table 1 do not appear as pairs. Were there pairs in which only one of them performed the tasks?

Indeed, the number of sessions were mostly not the same for a pair. This is explained by the different types of analysis performed in Figure 3 and for Table 1. For figure 3 we have taken into account any day that an animal had access to the device independent of whether a single interaction with the device was performed. In contrast, table 1 presents the number of sessions as days in which an animal had access to the device and performed trials, i.e. number of sessions in which learning could have occurred and data on a task were gathered. On some days only one of the animals of a pair interacted with the device and thus this animal would have a different session count than its cage mate. Additionally, sometimes one animal might not have gotten access to the device due to husbandry, veterinary or other experimental reasons. Together, this explains why the number of sessions are correlated for a pair but are not the same. In the updated table we now specify the cage mate of each animal.

(6) Lines 682-685 “Here, animals were observed to weakly reach, ...” Here describes failure of sensor detection lead animals to touch other objects as an alternative strategy of marmosets. If a sensor to detect animal’s action is not reliable, is it better to provide visual feedback for correct touches detected by touch pad sensor (e.g. flash in the surrounding of visual target) in addition to a reward?

We would like to thank the reviewer for this suggestion. Originally we did not implement a separate ‘correct touch’ feedback as in our task versions the screen always changes if the animal touches the screen successfully and thus implicitly provides visual feedback of correct touches. This, we agree with the reviewer, might not be enough. For future task versions where e.g. touch-and-hold is required from animals we are therefore considering implementing a visual feedback for acquired touch.

(7) Lines 330-331 “along the horizontal upper midline of the screen” What is meant here is not clear.

We agree that this sentence was hard to follow. The stimuli were moved between steps from the centre of the screen to the sides along an imaginary horizontal axis located 7 cm lower than the top of the screen.

As we have completely rewritten this section the sentence is no longer in the manuscript.

(8) Line 428 typo “animals animals”

We have deleted the doubling in the caption for table 1.

REVIEWER COMMENTS

Reviewer #1 (Remarks to the Author):

The authors have extensively addressed the comments of all reviewers. Nevertheless, I still do not consider that this study provides a major advancement for testing auditory cognition in nonhuman primates. There have been various studies employing different behavioral techniques (2 AFC, go/nogo, habituation/dishabituation, incentivised) which have demonstrated various auditory cognitive abilities in primates, e. g., concept formation, working memory, and categorization. Thus I follow reviewer 2's comment that the adoption of an automated cage-based training system from visual to auditory stimuli provides only an incremental advancement that is 'not strong enough to be published in Nature Communication'.

Reviewer #2 (Remarks to the Author):

Similar to my previous review comments, the device Calapai et al. developed would not be substantially novel although probably many marmoset researchers appreciate their effort to elaborate this system. In this revised version, they emphasized the fact that this system is useful to train monkeys in the auditory task, especially the two-alternative forced-choice task (2AFC). Based on their system's key features (auditory training system, an automated, unsupervised training method), this reframing would make this paper more attractive.

To further validate the utility of their system on auditory cognitive tests, they performed a new experiment to measure the auditory detection threshold under a naturalistic sound environment. However, perhaps due to the training history of preceding tasks, this task is not well-designed to simply detect the auditory threshold. In fact, what monkeys were required were: 1) if no sound, pick triangles, 2) if a pure tone was played, pick triangles, 3) if a vocalization was played, pick a face picture. Therefore, results are difficult to interpret. Can the authors plot psychometric functions for pure tone trials? Also, for the 0 dB stimulus in Fig 5B, do monkeys select triangles in most of the trials? A more simple task would be better to quantify the auditory threshold: e.g., if no sound, pick triangles. If sound, pick other pictures.

Minor comments

Figure 2A: Are there any reasons why hit rates for the specific steps are lower?

There are two figure 3s.

Page 12, line 217. Instead of using “Acoustic discrimination”, “Audio-visual association” would be an accurate label for this step. In fact, this step requires not only discriminating the auditory stimulus but also associating it with a visual counterpart and inhibiting response to another visual distractor.

Reviewer #3 (Remarks to the Author):

Authors re-structured the manuscript properly, even though it shows two versions of Figure 3. Besides that, the revised manuscript looks much better than the original. Additional behavioral data suggests rather that the system developed by the authors could be promising.

I have a few concerns.

(1) The manuscript includes two Figure 3s. It is probably just a mistake that authors did not check the document before finalizing submission. However, some descriptions related Figure 3 in main texts are inconsistent with both Figure 3s: e.g., Lines 804-807.

(2) The factors (independent variables) of Kruskal-Wallis tests are not clear. Detailed statistics should be reported. What are the criterion levels of significance for Kruskal Wallis and post hoc?

(3) Plots of reaction time in Figure 3 change the width of bars. What is the meaning of those widths?

(4) Figure 5B. Solid black curves are probably fitted. Please describe the expression the curves. Are there error bars or confidence intervals for data points?

(5) How were sound levels measured?

(6) Lines 300-302 describe “The estimated hearing threshold for the vocalization stimulus (mean 41.3 dB SPL; 39 for animal a, 48 for animal b, 37 for animal d) was below the background noise of the facility of 60 dB SPL (Figure 5C).”

However, plot of Figure 5C is expressed as attenuation level that is different from x-axis of Figure 5B. So, it probably needs more explanation.

(7) Line 613, “whom where” should be probably “that were”

(8) Line 404, “see e.g. (Hirokawa” should be “(see e.g. Hirokawa”

REVIEWER COMMENTS

Reviewer #1 (Remarks to the Author):

The authors have extensively addressed the comments of all reviewers. Nevertheless, I still do not consider that this study provides a major advancement for testing auditory cognition in nonhuman primates. There have been various studies employing different behavioral techniques (2 AFC, go/nogo, habituation/dishabituation, incentivised) which have demonstrated various auditory cognitive abilities in primates, e. g., concept formation, working memory, and categorization. Thus I follow reviewer 2's comment that the adoption of an automated cage-based training system from visual to auditory stimuli provides only an incremental advancement that is 'not strong enough to be published in Nature Communication'.

We would like to thank the reviewer for the appreciation of our work and the comments that helped us to further improve our manuscript.

We believe that there are several advancements in our study that have major implications. First, regarding the adoption of our automated approach, we have admittedly used the term "automated" somewhat loosely and we have rephrased sections of the main text to better reflect the implications we think the automated approach yields from scientific, practical, and ethical points of view.

Second, while it is true that auditory cognition in non-human primates has been studied in various publications, to the best of our knowledge our study is the first one to demonstrate flexible auditory cognition in unrestrained marmosets directly in their home enclosure. In this respect we also provide specific suggestions on how to effectively train marmosets to operate various audio-visual associations, without relying on fluid/food control or social separation, that we think is unprecedented.

Following the suggestion of reviewer 2 we have now reframed our manuscript highlighting the key features to make the paper more attractive.

Reviewer #2 (Remarks to the Author):

We would like to thank the reviewer for the appreciation of our work and the comments that helped us to further improve our manuscript.

Similar to my previous review comments, the device Calapai et al. developed would not be substantially novel although probably many marmoset researchers appreciate their effort to elaborate this system. In this revised version, they emphasized the fact that this system is useful to train monkeys in the auditory task, especially the two-alternative forced-choice task (2AFC). Based on their system's key features (auditory training system, an automated, unsupervised training method), this reframing would make this paper more attractive.

We would like to thank the reviewer for suggesting to reframe the manuscript. Now, we have rephrased parts of the manuscript to more accurately reflect the key features listed by reviewer #2, namely the auditory training system, and the automated unsupervised training method, with the addition of the self-paced, unrestrained, approach that does not rely on

fluid/food control nor social isolation.

To further validate the utility of their system on auditory cognitive tests, they performed a new experiment to measure the auditory detection threshold under a naturalistic sound environment. However, perhaps due to the training history of preceding tasks, this task is not well-designed to simply detect the auditory threshold. In fact, what monkeys were required were: 1) if no sound, pick triangles, 2) if a pure tone was played, pick triangles, 3) if a vocalization was played, pick a face picture.

Therefore, results are difficult to interpret. Can the authors plot psychometric functions for pure tone trials?

The reviewer is right in pointing out that the task structure cannot easily be used to interpret the psychometric functions to reveal auditory thresholds. When we prepared the figures for the manuscript we also plotted psychometric functions for the pure tone trials. As expected from the task design these were flat (see below). Therefore, we were convinced that the monkeys did perform a vocalization detection task allowing to assess auditory thresholds. In the revised manuscript, we have now taken up the suggestion of the reviewer and repeated these experiments with a simpler task design (see next paragraph).

Also, for the 0 dB stimulus in Fig 5B, do monkeys select triangles in most of the trials? A more simple task would be better to quantify the auditory threshold: e.g., if no sound, pick triangles. If sound, pick other pictures.

We would like to thank the reviewer for pointing out that a simpler task would be more appropriate to quantify auditory thresholds. Indeed, the original task in the new experiment followed from the training history of the monkeys.

In the revised manuscript we have now assessed auditory thresholds based on the task the reviewer suggested: monkeys had to pick the triangles if no sound was presented and if a sound was presented the monkeys had to choose the picture. With this new dataset we have replaced the psychophysical data in figure 5.

Minor comments

Figure 2A: Are there any reasons why hit rates for the specific steps are lower?

Although we designed the automated unsupervised training protocol to smoothly train the animals from naive to expert, certain steps might be more difficult than others to overcome. More difficult steps require more trials and thus the hit rate of those steps is lower compared to easier steps. We have added this information in the main text (Lines 207 - 210) to more clearly explain the relationship between hit rate and task difficulty in the automated unsupervised training protocol.

There are two figure 3s.

We apologize for this mishap. This issue might have arisen during the automatic conversion of our word documents to PDFs during the submission procedure. In the automatically created PDF file the second figure 3 was the correct one.

Page 12, line 217. Instead of using “Acoustic discrimination”, “Audio-visual association” would be an accurate label for this step. In fact, this step requires not only discriminating the auditory stimulus but also associating it with a visual counterpart and inhibiting response to another visual distractor.

We took up the reviewers’ suggestion to replace “acoustic discrimination” with “Audio-visual association” throughout the manuscript to better reflect the nature of the task.

Reviewer #3 (Remarks to the Author):

Authors re-structured the manuscript properly, even though it shows two versions of Figure 3. Besides that, the revised manuscript looks much better than the original. Additional behavioral data suggests rather that the system developed by the authors could be

promising.

We would like to thank the reviewer for the appreciation of our work and the comments that helped us to further improve our manuscript.

I have a few concerns.

(1) The manuscript includes two Figure 3s. It is probably just a mistake that authors did not check the document before finalizing submission. However, some descriptions related Figure 3 in main texts are inconsistent with both Figure 3s: e.g., Lines 804-807.

We apologize for this mishap. This issue might have arisen during the automatic conversion of our word documents to PDFs during the submission procedure. At the time of submission several attempts of downloading the automatically created PDF failed. This prevented us for catching this and the strange formatting of pages 12 - 20. In the revised manuscript we further adjusted the figure panel references such that the descriptions are now consistent.

(2) The factors (independent variables) of Kruskal-Wallis tests are not clear. Detailed statistics should be reported. What are the criterion levels of significance for Kruskal Wallis and post hoc?

We have added this information to the main text (see Methods - Data treatment). Generally, we considered an alpha level of less than 0.05 to be significant. In case of multiple comparisons we adjusted the alpha level with a Bonferroni correction.

(3) Plots of reaction time in Figure 3 change the width of bars. What is the meaning of those widths?

Like more conventional Boxplots, the Letter Value plots (*Boxenplot* in the *seaborn* python library used to create our plots) are used to represent large non-parametric distributions. While Boxplots represent the standard Q1 to Q3 quantile with a single solid box, in Letter Value plots additional boxes are used to represent more quantiles and visualize different partitions of a dataset. In Letter Value plots, the box following the central one represents the 25% percentile and the one following it represents the 12,5%. Each additional box represents percentiles containing half of the data of the previous percentile. We chose this representation because it more accurately represents information of data points in the tails of the distribution. We find this particularly useful for response time data which commonly have an exponential-like distribution.

(4) Figure 5B. Solid black curves are probably fitted. Please describe the expression the curves. Are there error bars or confidence intervals for data points?

We added a description of the psychometric function estimation in the methods section (see Methods - Data treatment). Based on the reviewers' suggestion we now provide confidence intervals for the obtained thresholds.

(5) How were sound levels measured?

A description of the procedure was added in the methods section (*Psychoacoustic assessments*). Briefly, MXBIs were placed inside a sound attenuating booth and a measurement microphone (Bruel And Kjaer Type 4966 1/2-inch) was placed at the marmoset ear level pointing towards one speaker. The microphone was connected to a measurement amplifier (Bruel And Kjaer Measuring Amplifiers Types 2610) to measure sound pressure levels.

(6) Lines 300-302 describe “The estimated hearing threshold for the vocalization stimulus (mean 41.3 dB SPL; 39 for animal a, 48 for animal b, 37 for animal d) was below the background noise of the facility of 60 dB SPL (Figure 5C).”

However, plot of Figure 5C is expressed as attenuation level that is different from x-axis of Figure 5B. So, it probably needs more explanation.

We have added more explanation of this analysis/measurement in the methods section. In short, figure 5C was meant to illustrate the average spectrum of the colony background rather than demonstrating the particular sound level. This panel would be normalized to the frequency resolution (db/Hz) even when converted into level in dB and therefore still be difficult to integrate with the sound pressure levels measured for the hearing thresholds. Because of this we stated the measured background noise level of the facility.

(7) Line 613, “whom where” should be probably “that were”

We agree and corrected the main text

(8) Line 404, “see e.g. (Hirokawa” should be “(see e.g. Hirokawa”

We corrected the main text.

REVIEWERS' COMMENTS

Reviewer #2 (Remarks to the Author):

Thank you for addressing my concerns. New results proved that the authors' system allows us to conduct auditory psychophysical experiments in the marmoset home cage. It would be difficult or take a long time to obtain such nice psychometric functions even by human-based training. I just have a minor suggestion. Since readers would see the MXBI system (Fig 6) and training design (Fig 7) at the end of this paper (Methods section), I would suggest putting those figures in the earlier part of the Results section and referring to each panel when needed.

Reviewer #3 (Remarks to the Author):

Line 450 bothers me. The way the word "e.g." is used in the line is unorthodox. Otherwise, my concerns are clarified, and I see no more issues.

RESPONSES TO REVIEWER COMMENTS

Reviewer #2 (Remarks to the Author):

Thank you for addressing my concerns. New results proved that the authors' system allows us to conduct auditory psychophysical experiments in the marmoset home cage. It would be difficult or take a long time to obtain such nice psychometric functions even by human-based training. I just have a minor suggestion. Since readers would see the MXBI system (Fig 6) and training design (Fig 7) at the end of this paper (Methods section), I would suggest putting those figures in the earlier part of the Results section and referring to each panel when needed.

We would like to thank the reviewer for the continuous critical evaluation of our work which helped to substantially improve the manuscript.

Based on the reviewer's suggestion to introduce the MXBI system earlier in the manuscript, we have opted to include parts of the figures 6 and 7 in figures 1, 2 and 3. Specifically, panels 6C, d were included as new panel 1A; panel 6E as new panel 2 B; panel 7 D included as parts of revised panels 3A, B. The remaining panels were put together as new supplementary figure S4. All panels were referenced as needed.

We hope that this allows to keep the main points of the device accessible to a general audience while providing more details for interested readers in the methods and supplement.

Reviewer #3 (Remarks to the Author):

Line 450 bothers me. The way the word "e.g." is used in the line is unorthodox. Otherwise, my concerns are clarified, and I see no more issues.

We would like to thank the reviewer for the continuous critical evaluation of our work which helped to substantially improve the manuscript.

Based on the reviewer's comment we have deleted the word 'e.g.' as it indeed was an odd way of referencing the previous sentence. The relevant part now reads:

"Also, the sound was presented from the speaker on which side the correct visual response indicator was located. This has been shown to be a strong cue for ferrets to guide choice towards the respective sound direction. "